# Presenilin mutations deregulate mitochondrial Ca$^{2+}$ homeostasis and metabolic activity causing neurodegeneration in *Caenorhabditis elegans*

Shaarika Sarasija[1], Jocelyn T Laboy[1], Zahra Ashkavand[1], Jennifer Bonner[2], Yi Tang[1], Kenneth R Norman[1]*

[1]Department of Regenerative and Cancer Cell Biology, Albany Medical College, Albany, United States; [2]Department of Biology, Skidmore College, Saratoga Springs, United States

**Abstract** Mitochondrial dysfunction and subsequent metabolic deregulation is observed in neurodegenerative diseases and aging. Mutations in the presenilin (PSEN) encoding genes (*PSEN1* and *PSEN2*) cause most cases of familial Alzheimer's disease (AD); however, the underlying mechanism of pathogenesis remains unclear. Here, we show that mutations in the *C. elegans* gene encoding a PSEN homolog, *sel-12* result in mitochondrial metabolic defects that promote neurodegeneration as a result of oxidative stress. In *sel-12* mutants, elevated endoplasmic reticulum (ER)-mitochondrial Ca$^{2+}$ signaling leads to an increase in mitochondrial Ca$^{2+}$ content which stimulates mitochondrial respiration resulting in an increase in mitochondrial superoxide production. By reducing ER Ca$^{2+}$ release, mitochondrial Ca$^{2+}$ uptake or mitochondrial superoxides in *sel-12* mutants, we demonstrate rescue of the mitochondrial metabolic defects and prevent neurodegeneration. These data suggest that mutations in PSEN alter mitochondrial metabolic function via ER to mitochondrial Ca$^{2+}$ signaling and provide insight for alternative targets for treating neurodegenerative diseases.

DOI: https://doi.org/10.7554/eLife.33052.001

*For correspondence:
normank@mail.amc.edu

**Competing interests:** The authors declare that no competing interests exist.

## Introduction

In all metazoans, mitochondria are essential organelles and mitochondrial dysfunction is frequently observed in neurodegenerative diseases and aging. While mitochondria have a well-established role in ATP generation, they also have a vital role in Ca$^{2+}$ homeostasis. Several studies have shown that release of endoplasmic reticulum (ER) Ca$^{2+}$ results in the elevation of mitochondrial Ca$^{2+}$ levels which in turn stimulates metabolic activity of the mitochondria (***Das and Harris, 1990***; ***Glancy and Balaban, 2012***; ***Hansford and Zorov, 1998***; ***McCormack and Denton, 1993***; ***Mildaziene et al., 1995***; ***Wernette et al., 1981***). Thus, insults or deregulated signaling between the ER and mitochondria can cause mitochondrial dysfunction and affect cellular fitness.

Alzheimer's disease (AD) is the leading cause of dementia in the elderly and accounts for between 60–80% of all cases of dementia. Familial Alzheimer's disease (FAD) is a subset of AD where there is a genetic predisposition to the disease as a result of mutations predominantly in the presenilin genes, *PSEN1* and *PSEN2* (***ALZFORUM, 2016***). PSENs are ~50 kDa multipass transmembrane proteins that primarily localize to the ER (***Bezprozvanny and Mattson, 2008***) and are enriched in the ER membrane associated with mitochondria (***Area-Gomez et al., 2009***). Notably, PSENs are the

**eLife digest** Alzheimer's disease is the most common type of dementia. A hallmark of this condition is progressive loss of memory, accompanied by a buildup of hard clumps of protein between the brain cells. These protein clumps, known as amyloid plaques, are a key focus of research into Alzheimer's disease. They are likely to be toxic to brain cells, but their role in the development and progression of the disease is not yet known.

Though the cause of Alzheimer's disease remains unclear, an inherited form of the disease may hold some clues. Mutations in genes for proteins called presenilins cause an earlier onset form of Alzheimer's disease, in which symptoms can develop in people who are in their 40s or 50s. The presenilin proteins appear in a cell structure called the endoplasmic reticulum, which plays many roles in the normal activities of a cell. Among other things, this structure stores and releases calcium ions, and cells use these ions to send and process many signals.

The cell's energy-producing powerhouses, the mitochondria, use calcium to boost their metabolic activity. This allows them to make more energy for the cell, but in the process they also make damaging byproducts. These byproducts include oxygen-containing chemicals, known as reactive oxygen species (ROS), which react strongly with other molecules. While low levels of ROS are a normal part of cell activity, if the levels get too high, these chemicals can attack and damage structures within the cell.

Untangling the effects of amyloid plaques and presenilins on brain cells in humans is challenging. But, a nematode worm called *Caenorhabditis elegans* does not form plaques, making it possible to look at presenilins on their own. Previous work in these worms has shown that presenilin mutations affect the endoplasmic reticulum and change the appearance of mitochondria. Here, Sarasija et al. extend this work to find out more about the effects presenilin mutations have on living cells.

Presenilin mutations in young adult worms increased the amount of calcium released by the endoplasmic reticulum. This increased the activity of the mitochondria and caused ROS levels to rise to damaging levels. This caused stress inside the cells, and the worms started to show early signs damage to their nervous systems. Mutations that decreased the movement of calcium from the endoplasmic reticulum to the mitochondria helped to prevent the damage. Treating the mitochondria with antioxidants to mop up the extra ROS also protected the cells.

This kind of damage to brain cells did not depend on amyloid plaques. Whilst the plaques are likely to be toxic, these new findings highlights the role that other chemical and biological processes might play in Alzheimer's disease. Further work to reveal the underlying cause of Alzheimer's disease may lead to new therapies to treat this condition in the future.

DOI: https://doi.org/10.7554/eLife.33052.002

critical aspartyl protease component of the gamma-secretase complex, a multi-subunit protease that resides in cellular membranes. Gamma-secretase is involved in the cleavage of amyloid precursor protein (APP) into beta amyloid (Abeta) peptides, and in the cleavage and activation of several other transmembrane proteins, including Notch (*Beel and Sanders, 2008*). Dominating the study of AD pathogenesis is the amyloid hypothesis which ascertains that altered PSEN function leads to an increase in the toxic Abeta42 peptide, and the eventual accumulation of amyloid plaques in the brain leads to neurodegeneration and dementia observed in AD patients (*Hardy, 2006*). However, it is noteworthy that while amyloid plaques serve as a histopathological hallmark of AD, there is no correlation between plaque load and the severity of dementia, and postmortem analyses of some AD patients with exaggerated cognitive decline have shown a lack of plaque formation (*Giannakopoulos et al., 2003*; *Guillozet et al., 2003*; *Terry et al., 1991*). Also, using neuroimaging techniques, extensive plaque formation has been observed in people with no cognitive impairment (*Nordberg, 2008*; *Villemagne et al., 2008*). Concomitantly, there is a large body of work that implicates exacerbated ER $Ca^{2+}$ release as the causative agent in AD pathogenesis (*Bandara et al., 2013*; *Chan et al., 2000*; *Cheung et al., 2008*; *Green et al., 2008*; *Leissring et al., 1999*; *Stutzmann et al., 2004*; *Tu et al., 2006*). Moreover, there is evidence of enhanced ER-mitochondria crosstalk in cells from FAD patients with mutations in *PSEN1*, *PSEN2*, and *APP*, as well as patients with sporadic AD (*Area-Gomez et al., 2012*) and enhanced ER to mitochondrial $Ca^{2+}$ transfer is

observed in cells expressing FAD-mutant PSEN2 (*Zampese et al., 2011*). Additionally, reactive oxygen-mediated oxidative damage has been observed in brains of AD patients (*Lovell and Markesbery, 2007*; *Lovell et al., 2011*) and an increase in hydrogen peroxide levels is observed in AD mice models even prior to the appearance of plaques (*Manczak et al., 2006*). However, despite decades of research, a clear connection between these seemingly disparate observations does not exist that explains the pathogenesis of AD.

In *C. elegans*, disruption of the PSEN ortholog, SEL-12, leads to ER $Ca^{2+}$ dysregulation that causes mitochondrial disorganization and reduced organismal health (*Sarasija and Norman, 2015*). Here, we examine whether these mitochondrial defects observed in *sel-12* mutants lead to mitochondrial metabolic defects that can result in neuronal dysfunction. From these analyses, we find that *sel-12* mutations result in increased mitochondrial $Ca^{2+}$ concentration, accelerated oxidative phosphorylation (OXPHOS), elevated reactive oxygen species (ROS) generation and oxidative stress mediated neurodegeneration. Remarkably, we demonstrate that reducing ER to mitochondrial $Ca^{2+}$ transfer in *sel-12* mutants can normalize mitochondrial $Ca^{2+}$ levels and function, reduce the levels of ROS and suppress neurodegeneration. Moreover, we demonstrate similar mitochondrial dysfunction in fibroblasts isolated from FAD patients. Lastly, we show that treating *sel-12* mutants with a mitochondria-targeted antioxidant suppresses the neurodegenerative phenotypes observed in *sel-12* mutants. Since *C. elegans* do not encode an Abeta peptide (*Daigle and Li, 1993*; *McColl et al., 2012*), our data indicates that neurodegeneration occurs in presenilin mutants by faulty ER to mitochondria $Ca^{2+}$ transfer leading to mitochondrial metabolic dysfunction, completely independent of Abeta signaling, and consummates in the rise of mitochondrial ROS generation to detrimental levels. Therefore, our study provides a comprehensive delineation of AD pathogenesis in an intact animal model and its importance is underscored by the fact that we provide evidence that Abeta signaling does not appear necessary for neurodegeneration.

## Results

### Mutations in *sel-12* increase mitochondrial matrix $Ca^{2+}$ concentration resulting in increased OXPHOS

Previously, we observed mitochondrial disorganization in the body wall muscle of *sel-12* animals, which could be rescued by reducing $Ca^{2+}$ release from the ER or mitochondrial $Ca^{2+}$ uptake (*Sarasija and Norman, 2015*). This led us to hypothesize that mutations in *sel-12* result in increased mitochondrial $Ca^{2+}$ levels. To test this hypothesis, we expressed in the body wall muscle, a mitochondrial matrix targeted GCaMP6 (a genetically encoded $Ca^{2+}$ indicator) along with a red fluorescent protein, mCherry, as an expression control (*Figure 1—figure supplement 1A*). We normalized the fluorescence intensity of the GCaMP6 to that of mCherry to determine the relative mitochondrial matrix $Ca^{2+}$ concentration in day one adult wild type animals and two *sel-12* mutants, *sel-12 (ar131)* and *sel-12(ty11)*. *sel-12(ar131)* mutants carry a missense mutation in the *sel-12* gene, which is a conserved change seen in FAD patients (*ALZFORUM, 2016*; *Levitan and Greenwald, 1995*) and *sel-12 (ty11)* mutants contain a premature stop codon in the *sel-12* gene, which is a predicted null mutation (*Cinar et al., 2001*). Strikingly, we observe that the relative fluorescence intensity of GCaMP6 to mCherry is 2.4 and 3.4 fold higher in *sel-12 (ar131)* and *sel-12(ty11)* animals, respectively, compared to wild type animals (*Figure 1A*). Therefore, consistent with our hypothesis, *sel-12* mutants have higher levels of mitochondrial matrix $Ca^{2+}$. Since *sel-12*, like *PSEN1*, is widely expressed, we examined mitochondrial $Ca^{2+}$ levels in a subset of neurons using the same ratiometric GCaMP6/mCherry construct. Similar to our observation of body wall muscle mitochondrial $Ca^{2+}$ levels, we detected higher fluorescence levels of GCaMP6 in the ALM and PLM light touch mechanosensory neurons of *sel-12* mutants consistent with elevated mitochondrial $Ca^{2+}$ levels in the nervous system (*Figure 1—figure supplement 1B*).

Although mitochondria can act as a $Ca^{2+}$ sink, they also respond dynamically to increases in $Ca^{2+}$ levels. Indeed, an increase in mitochondrial matrix $Ca^{2+}$ levels results in the coordinated up regulation of the tricarboxylic acid (TCA) cycle and OXPHOS, increasing ATP output and ROS levels (*Das and Harris, 1990*; *Glancy and Balaban, 2012*; *Hansford and Zorov, 1998*; *McCormack and Denton, 1993*; *Mildaziene et al., 1995*; *Wernette et al., 1981*). Thus, to investigate whether the elevated $Ca^{2+}$ levels observed in *sel-12* mutant mitochondria influence mitochondrial activity, we first

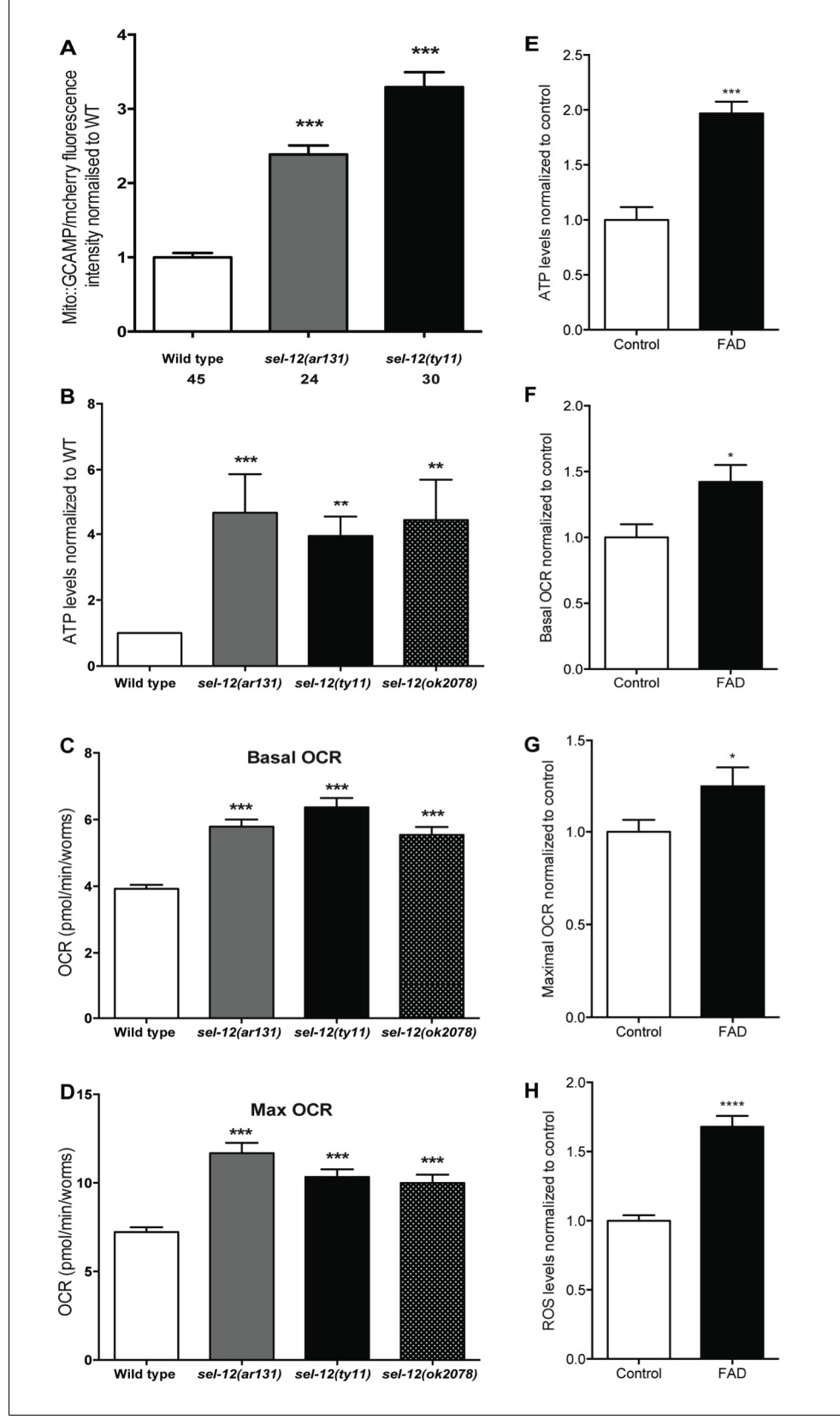

**Figure 1.** Mutations in *sel-12* and PSEN1 result in higher rates of oxygen consumption and generate higher levels of ATP. (**A**) Quantification of mitochondrial Ca$^{2+}$ using animals expressing mito::GCaMP6 and mCherry in their

*Figure 1 continued on next page*

*Figure 1 continued*

body wall muscle (*takEx347*). (**B**) Quantification of the relative ATP levels in *sel-12* mutants compared to wild type animals. Data are from three replicate assays. (**C**) Basal respiration rates of wild type and *sel-12* mutant animals. (**D**) Maximal respiration rates of wild type and *sel-12* mutant animals after exposure to FCCP. OCR data are from three replicate assays. (**E**) ATP levels normalized to protein content in skin fibroblasts isolated from control and FAD patients. (**F**) Basal and (**G**) Maximal OCR normalized to protein content in skin fibroblasts isolated from control and FAD patients. (**H**). ROS levels normalized to protein content in skin fibroblasts isolated from control and FAD patients. Data are displayed as mean ± SEM, and all comparisons have been made to wild type animals unless otherwise indicated. *$p < 0.05$, **$p < 0.01$, ***$p < 0.0001$, ****$p < 0.00001$ (One way. ANOVA with Tukey test for A-D, two-tailed T-test for E-H).

DOI: https://doi.org/10.7554/eLife.33052.017

The following source data and figure supplements are available for figure 1:

**Source data 1.** Raw data for *Figure 1*.
DOI: https://doi.org/10.7554/eLife.33052.019
**Figure supplement 1.** Mitochondrial $Ca^{2+}$ levels are higher in *sel-12* mutants.
DOI: https://doi.org/10.7554/eLife.33052.018
**Figure supplement 1—source data 1.** Raw data for *Figure 1—figure supplement 1*.
DOI: https://doi.org/10.7554/eLife.33052.020

analyzed ATP levels as a general read out of mitochondrial function. Remarkably, we observed that ATP levels are higher in *sel-12(ar131)*, *sel-12(ty11)* and *sel-12(ok2078)* (an additional null allele) animals when compared to age matched day one wild type animals (*Figure 1B*). Next, to determine whether this elevation in ATP levels was a result of amplified OXPHOS, we examined mitochondrial respiration by measuring the oxygen consumption rate (OCR) in *sel-12* mutant and wild type animals using a Seahorse XFp Analyzer. While wild type animals had a basal OCR of 3.9 pmol/min/worm, *sel-12(ar131)*, *sel-12(ty11)* and *sel-12(ok2078)* animals respired at a rate of 5.8, 6.4 and 5.5 pmol/min/worm, respectively (*Figure 1C*). Similarly, when challenged with carbonyl cyanide 4-(trifluoromethoxy) phenylhydrazone (FCCP), an uncoupler of mitochondrial OXPHOS, we detected an increased maximal OCR of 11.7, 10.3 and 10 pmol/min/worm in *sel-12(ar131)*, *sel-12(ty11)* and *sel-12(ok2078)* animals respectively when compared to the maximal OCR of 7.2 pmol/min/worm in wild type animals (*Figure 1D*). These data indicate that *sel-12* mutants have elevated OXPHOS activity, which results in higher OCR and ATP levels. Consistent with elevated mitochondrial activity, we have previously reported high ROS levels in *sel-12* mutants (*Sarasija and Norman, 2015*). Interestingly, as *sel-12* mutants age, mitochondrial respiration dramatically decreases compared to wild type animals suggesting a reduction of mitochondrial function in old animals (*Figure 1—figure supplement 1C, D*).

The elevated levels of ATP, ROS and mitochondrial $Ca^{2+}$ as well as the high OCR observed in young adult *sel-12* mutants could arise due to increased mitochondrial content in *sel-12* mutants. Thus, to investigate this possibility, we first used qRT-PCR to measure relative mitochondrial DNA (mitoDNA) copy number versus nuclear DNA in day one adult wild type and *sel-12* mutants. We found that the relative mitoDNA copy number in *sel-12* mutants is similar to wild type animals (*Figure 1—figure supplement 1E*). Additionally, we measured fluorescence intensity of day one adult animals expressing a TOM20::GFP integrated transgene that labels the outer mitochondrial membrane in the body wall muscle. We found that fluorescence intensity between wild type and *sel-12* mutants are indistinguishable (*Figure 1—figure supplement 1F*). These data suggest that the changes in ATP and mitochondrial $Ca^{2+}$ levels and OCR are not caused by an increase in mitochondrial biogenesis or mitochondrial accumulation.

## Skin fibroblasts isolated from FAD patients have increased ATP, OCR and ROS levels

A recent study on astrocytes differentiated from induced pluripotent stem cells (iPSCs) derived from AD patients with *PSEN1* mutations demonstrated high OCR and ROS levels in these cells, which could be rescued by CRISPR/Cas9 correction of the mutation (*Oksanen et al., 2017*). These data along with our data suggest a conserved role of PSEN1 and SEL-12 in regulating mitochondrial activity. To investigate this further, we analyzed mitochondrial activity in skin fibroblast cells isolated from

FAD patients with PSEN1 mutations. First, we examined ATP levels to obtain an approximation of mitochondrial activity in control and FAD fibroblasts. As we observed in *sel-12* mutants, we found that ATP levels are significantly higher in FAD fibroblasts compared to control fibroblasts (*Figure 1E*). Next, we investigated mitochondrial respiratory rate by measuring oxygen consumption in FAD and control fibroblasts. Again, consistent with the *sel-12* mutants and similar to FAD astrocytes (*Oksanen et al., 2017*), we found that the basal and maximal OCRs are significantly higher in FAD fibroblasts compared to control fibroblasts (*Figure 1F,G*). Lastly, since we previously found high levels of ROS in *sel-12* mutants (*Sarasija and Norman, 2015*) and ROS is produced by mitochondrial respiratory activity, we examined ROS levels in FAD and control fibroblasts. Like *sel-12* mutants and FAD astrocytes (*Oksanen et al., 2017*), we found elevated ROS in the FAD fibroblasts compared to control fibroblasts (*Figure 1H*). These observations are consistent with PSEN1 having a role in mitochondrial activity in human fibroblasts similar to the role SEL-12 has in *C. elegans*.

## *sel-12* mutants have disorganized mitochondrial morphology in mechanosensory neurons

In addition to body wall muscle mitochondria, we previously observed mitochondrial disorganization in a subset of interneurons in *sel-12* mutants (*Sarasija and Norman, 2015*) and therefore, sought to determine whether mutations in *sel-12* lead to mitochondrial disorganization in other neuronal cell types and, if so, investigate whether this leads to a physiological defect in neuronal function in vivo. We focused on touch receptor neurons, a class of mechanosensory neurons that respond to light touch. Light touch to the body of *C. elegans* is sensed by six mechanosensory neurons (ALML/R, PLML/R, AVM, and PVM) whose structural and functional neurodegeneration has been well-characterized (*Pan et al., 2011*; *Tank et al., 2011*; *Toth et al., 2012*). Using transgenic animals that express mCherry targeted to the outer mitochondrial membrane in mechanosensory neurons (*Hsu et al., 2014*), we compared the mitochondrial structure in wild type animals, *sel-12(ar131)* and *sel-12(ty11)* at day one of adulthood. Unlike wild type animals, which maintain their mitochondria mostly in a connected network (*Figure 2A*) (*Hsu et al., 2014*), we found that 53.3 and 55% of *sel-12 (ar131)* and *sel-12(ty11)* mutants, respectively, have discontinuous morphology of ALM neuronal mitochondria compared to the 16.7% of wild type animals with discontinuous mitochondria morphology (*Figure 2A*). This data suggests that SEL-12 function is required to maintain mitochondrial structure and is consistent with our previous observations of mitochondrial disorganization in the body wall muscle and interneurons of *sel-12* mutants (*Sarasija and Norman, 2015*).

## SEL-12 is required for mechanosensation

Given the extent of structural disorganization observed in the mechanosensory neuronal mitochondria of *sel-12* mutants, we decided to investigate whether *sel-12* mutations lead to a behavioral defect, and examined light touch response in these animals. A freely crawling animal when touched by an eyebrow hair just posterior of the pharynx (anterior touch) responds normally with a reversal in their motion and when touched slightly anterior of the anus (posterior touch), will move forward; a response that dampens as the animal ages into mid-late adulthood (*Pan et al., 2011*; *Tank et al., 2011*; *Toth et al., 2012*). We examined the response to anterior and posterior light touch in *sel-12 (ar131), sel-12(ty11),* and *sel-12(ok2078)* at L4 larval stage (last larval stage before adulthood), and days 1 and 3 of adulthood. From these analyses, we found that *sel-12* mutants respond similarly to wild type animals in their L4 larval stage but show a significant reduction in the response to light touch at day 1 of adulthood with only 45.1, 39 and 39.2% positive response in *sel-12 (ar131), sel-12 (ty11)* and *sel-12 (ok2078)* mutants, respectively, compared to the 83.1% positive response in wild type animals (*Figure 2B*). Moreover, this response to light touch progressively worsens when this behavior is examined in day three adults (*Figure 2B*). In order to determine whether the mechanosensation defect in adult animals is due to the loss of SEL-12 function in the nervous system, we investigated mechanosensation in *sel-12(ty11)* null mutants expressing wild type SEL-12 under a pan-neuronal or muscle-specific promoter (*Sarasija and Norman, 2015*). Pan-neuronal expression of wild type SEL-12 rescues the mechanosensory defects in *sel-12(ty11)* mutants while the expression of wild type SEL-12 under a muscle-specific promoter fails to rescue (*Figure 2C*).

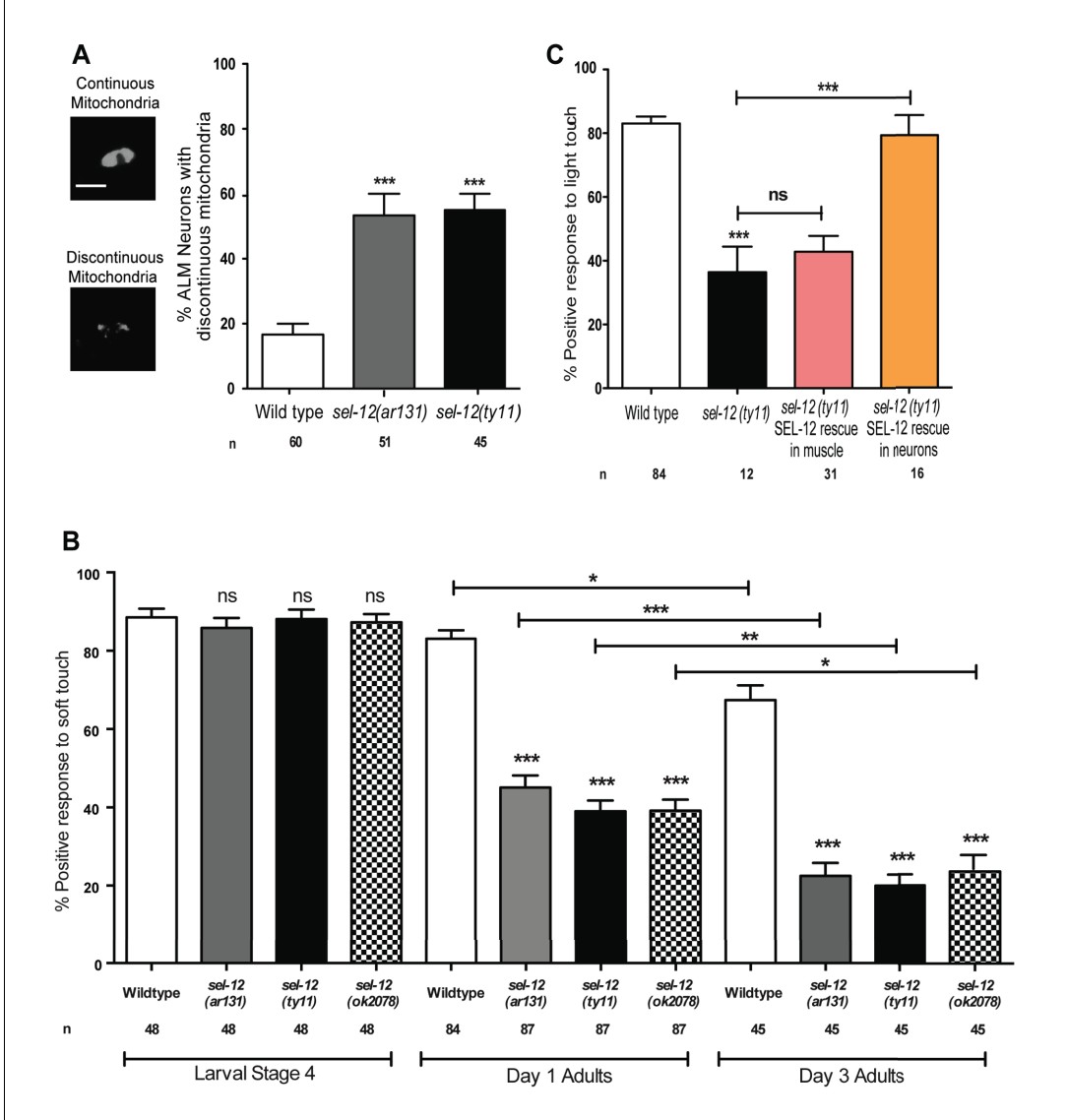

**Figure 2.** SEL-12 is required for mitochondrial structural maintenance and mechanosensation. (**A**) Representative images and quantification of the incidence of discontinuous ALM neuronal mitochondria. Analysis was done using transgenic animals expressing mCherry fused with the outer mitochondrial membrane protein, TOMM-20 in mechanosensory neurons (*twnEx8*). Scale bar represents 10 µm. (**B**) Response of wild type and *sel-12* mutants to anterior and posterior light touch at larval stage 4, day 1 and 3 of adulthood. (**C**) Response of day one *sel-12(ty11)* animals with tissue specific SEL-12 expression to light touch. n = number of animals analyzed per genotype. Data are displayed as mean ± SEM, and all comparisons have been made to wild type animals unless otherwise indicated. ns p>0.05, *p<0.05, **p<0.001, ***p<0.0001 (One way ANOVA with Tukey test).
DOI: https://doi.org/10.7554/eLife.33052.003

The following source data is available for figure 2:

**Source data 1.** Raw data for *Figure 2*.
DOI: https://doi.org/10.7554/eLife.33052.004

## Mechanosensory neurons exhibit structural neurodegeneration in *sel-12* mutants

Since we have found mitochondrial disorganization in the mechanosensory neurons (*Figure 2A*) and mechanosensory defects (*Figure 2B*) in adult *sel-12* mutants, we next examined the morphology of these neurons for any signs of neurodegeneration. In healthy animals, ALM neurons comprise a small circular soma, a long anterior process that extends into the nerve ring and an occasional short posterior process with no defined function (*Figure 3A* top, middle). Previously, it has been shown that as

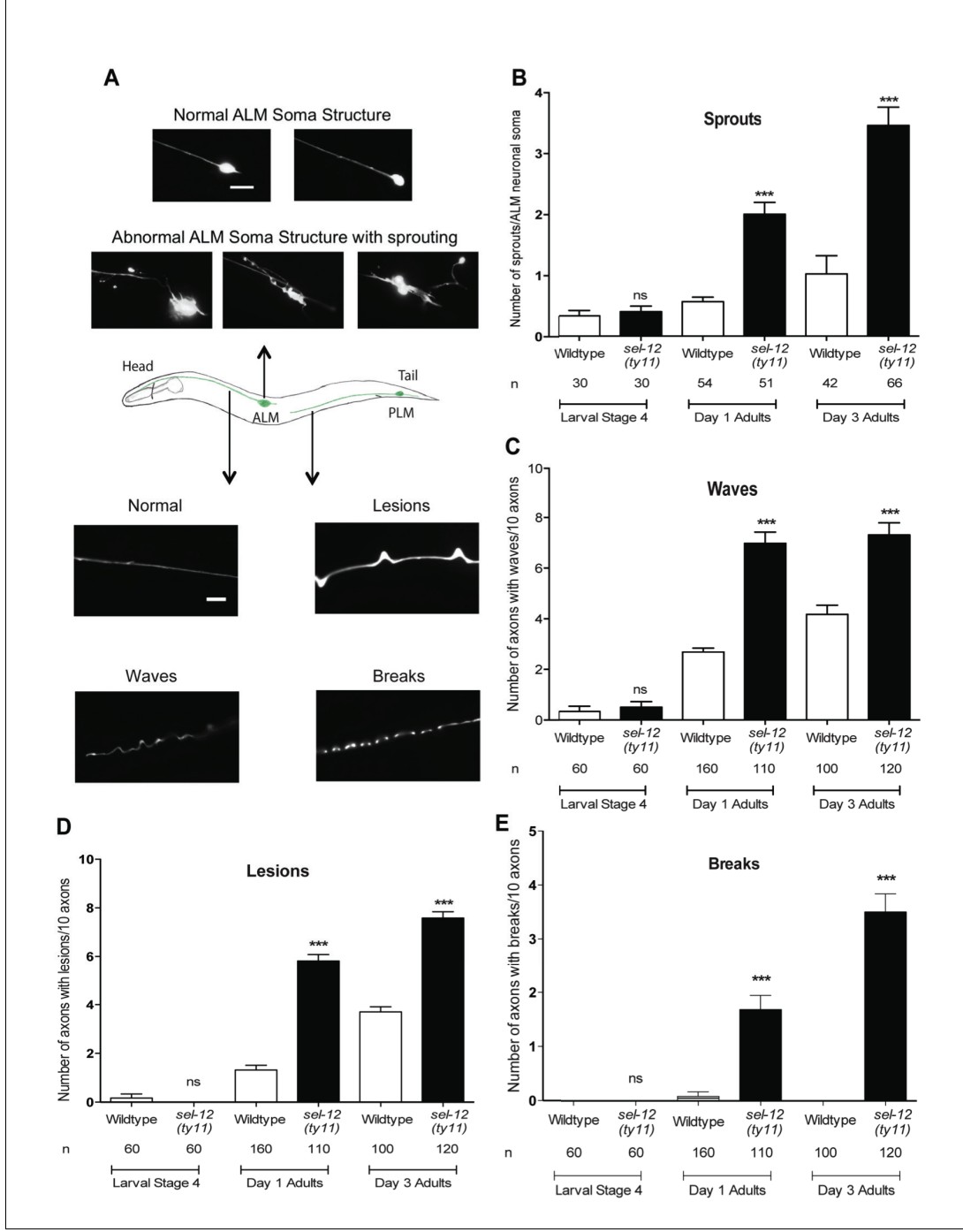

**Figure 3.** Loss of SEL-12 function results in neurodegeneration. (**A**) Representative images of normal ALM neuronal soma and abnormal soma with ectopic sprouting (above), cartoon depicting location of ALM and PLM neuron in *C. elegans* (middle) and representative images of wave-like processes, lesions and breaks observed in the ALM and PLM neuronal processes (below). Scale bar represents 10 μm. (**B**) Quantification of aberrant structures (sprouts/branches) on ALM neuronal soma at larval stage 4, day 1 and 3 of adulthood. Quantification of the frequency of wave-like processes (**C**), lesions (**D**) and breaks (**E**) in ALM and PLM neuronal processes in wild type and *sel-12(ty11)* animals at larval stage 4, day 1 and 3 of adulthood. Neuronal morphology analysis is done using transgenic animals expressing *mec-4p*::GFP (*zdIs5*). n = number of animals analyzed per genotype. Data are displayed as mean ± SEM, and all comparisons have been made to wild type animals unless otherwise indicated. ***p<0.0001 (One way ANOVA with Tukey test).

DOI: https://doi.org/10.7554/eLife.33052.005

*Figure 3 continued on next page*

*Figure 3 continued*

The following source data and figure supplements are available for figure 3:

**Source data 1.** Raw data for *Figure 3*.
DOI: https://doi.org/10.7554/eLife.33052.007
**Figure supplement 1.** SEL-12 has a tissue-specific but protease independent role in neuronal health maintenance.
DOI: https://doi.org/10.7554/eLife.33052.006
**Figure supplement 1—source data 1.** Raw data for *Figure 3—figure supplement 1*.
DOI: https://doi.org/10.7554/eLife.33052.008

adult wild type animals age they show ectopic branches sprout from the ALM neuronal soma, which is considered to be a sign of age associated neurodegeneration (*Pan et al., 2011*; *Tank et al., 2011*; *Toth et al., 2012*). While ALM in L4 larval stage *sel-12* mutants is indistinguishable from wild type animals, strikingly *sel-12* mutants show ectopic branches emanating from the soma as adults (*Figure 3A* top, *Figure 3B*). Day one *sel-12(ty11)* adults show on average two sprouts/ALM soma compared to 0.56 sprouts/ALM soma in day one wild type adults (*Figure 3A,B*). Furthermore, as they age further, the *sel-12(ty11)* animals develop upwards of 3 sprouts/ALM soma (while age matched wild type animals only have one sprout/ALM soma) (*Figure 3A,B*), paralleling the loss in mechanosensation (*Figure 2B*). Also, supporting a neuronal tissue specific role of SEL-12, we found that pan-neuronal expression of wild type SEL-12 decreases the sprouting in adult *sel-12(ty11)* animals from 2.8 sprouts/ALM soma to 1.2 sprouts/ALM soma while muscle specific expression of SEL-12 does not (*Figure 3—figure supplement 1A*).

Along with the structural aberrations observed in the ALM soma, the axonal processes of mechanosensory neurons are known to undergo age-associated neurodegeneration (*Pan et al., 2011*; *Tank et al., 2011*; *Toth et al., 2012*). Strikingly, we observe morphological abnormalities in the axonal processes of the mechanosensory neurons in adult *sel-12(ty11)* animals at a higher frequency than in age-matched wild type animals. The ALM and PLM axonal processes of *sel-12(ty11)* animals at day 1 of adulthood develop a wavy appearance at a frequency of ~7 per 10 ALM/PLM processes while age-matched wild type animals show only ~2 per 10 ALM/PLM processes (*Figure 3A* bottom, *Figure 3C*). The appearance of irregular lesions that appear to give way to breaks in the neuronal processes is also noted. Day one adult *sel-12(ty11)* animals have irregular lesions (*Figure 3A* bottom, *Figure 3D*) and breaks in their neuronal processes (*Figure 3A* bottom, *Figure 3E*) at a rate of ~6 and ~2 per 10 ALM/PLM processes, respectively compared to wild type animals at ~1 and ~0 per 10 ALM/PLM processes, respectively. *sel-12* mutants appear normal at the L4 larval stage but as they enter adulthood, their propensity of developing abnormal processes, lesions and breaks also increases (*Figure 3C–E*) similar to our observation with ectopic sprouting (*Figure 3B*). Taken together, the neuronal structural aberrations and decline in mechanosensory perception in adult *sel-12* mutants appear to be structural and functional indicators of neurodegeneration in these animals rather than symptoms of developmental defects.

## The mechanosensory neuronal defects observed in *sel-12* mutants are independent of Notch signaling and gamma-secretase activity

Presenilin is the catalytic component of the gamma-secretase complex that is responsible for proteolytic cleavage and activation of Notch signaling. Due to the pervasive nature of Notch signaling, we investigated whether the mechanosensory neuronal defects observed in *sel-12* mutants are the result of Notch signaling loss. *C. elegans* possess two genes encoding Notch orthologs, *lin-12* and *glp-1*. We assayed mechanosensation in both *lin-12* and *glp-1* mutants to determine if loss of Notch signaling was sufficient to cause mechanosensory defects akin to *sel-12* mutants. Unlike *sel-12* mutants, both *lin-12 and glp-1* mutants displayed responses to light touch at 86.4 and 78.5% as day one adults similar to the 83.1% positive response in day one aged wild type animals (*Figure 4A*). Moreover, the light touch response of *lin-12* and *glp-1* mutants was indistinguishable from wild type animals even at day 2 and day 3 of adulthood (*Figure 4A*), suggesting that Notch signaling is not required for mechanosensation.

Gamma-secretase activity is also involved in the cleavage of several other transmembrane proteins, which includes, but is not limited to, APP (*Beel and Sanders, 2008*). Thus, we next

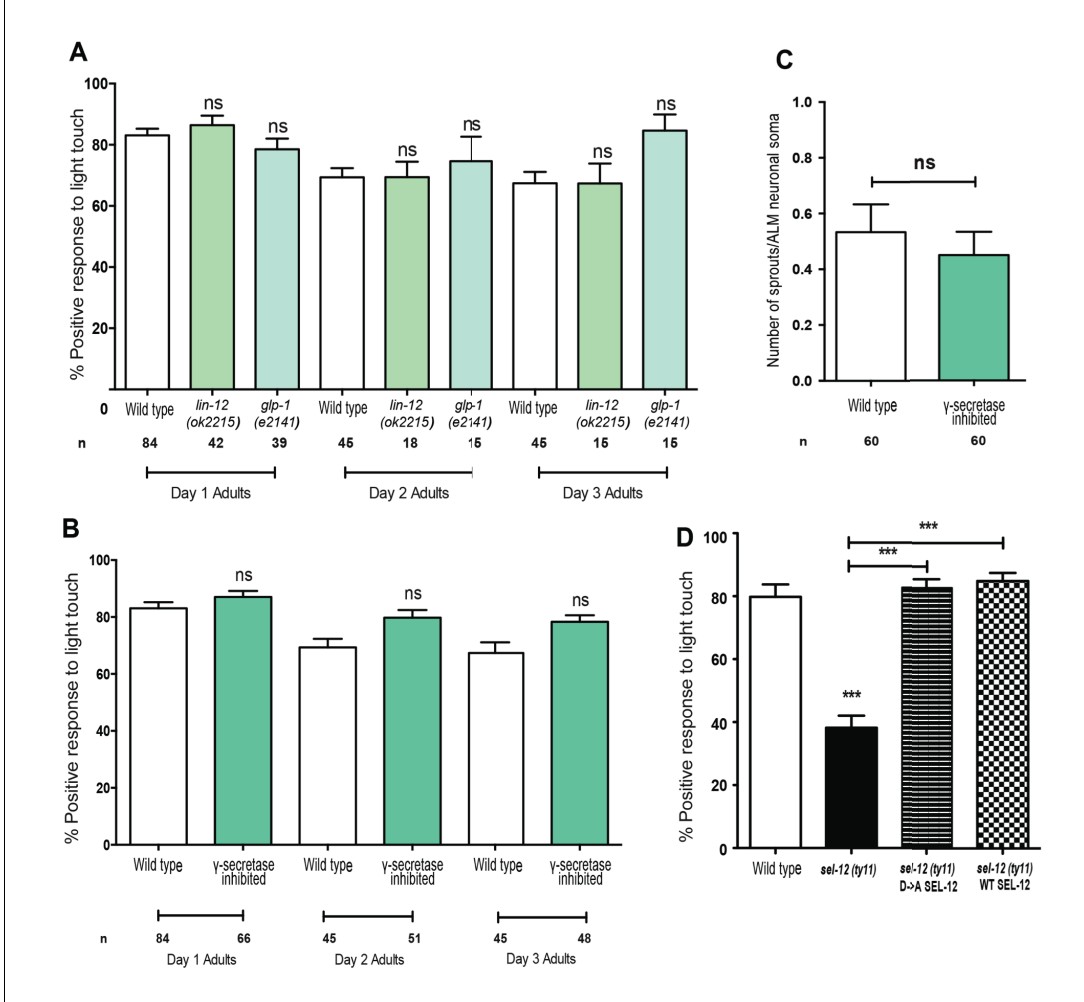

**Figure 4.** Notch signaling and gamma-secretase activity is not required for mechanosensation or mechanosensory neuron morphology. (A) Quantification of the response of Notch mutants (*glp-1* and *lin-12* mutants) to light anterior and posterior touch at day 1, 2 and 3 of adulthood. (B) Quantification of response to light touch in animals treated with gamma-secretase inhibitor (Compound E). (C) Quantification of aberrant neuronal structures (sprouts/branches) in ALM neurons in day one adult animals treated with gamma-secretase inhibitor (Compound E). (D) Quantification of the response of wild type, *sel-12(ty11)* and *sel-12(ty11)* animals expressing wild type or a protease dead SEL-12 to anterior and posterior light touch. ALM neuronal morphology analysis is done using transgenic animals expressing *mec-4p*::GFP (*zdIs5*). n = number of animals examined per genotype. Data are displayed as mean ± SEM, and all comparisons have been made to wild type animals unless otherwise indicated. ns p>0.05, ***p<0.0001 (One way ANOVA with Tukey test for A,B, and D, two-tailed T-test for C).

DOI: https://doi.org/10.7554/eLife.33052.009

The following source data and figure supplements are available for figure 4:

**Source data 1.** Raw data for *Figure 4*.
DOI: https://doi.org/10.7554/eLife.33052.011

**Figure supplement 1.** Loss of gamma-secretase function does not result in neurodegenerative morphologies.
DOI: https://doi.org/10.7554/eLife.33052.010

**Figure supplement 1—source data 1.** Raw data for *Figure 4—figure supplement 1*.
DOI: https://doi.org/10.7554/eLife.33052.012

investigated whether mechanosensation is mediated in a gamma-secretase dependent but Notch-independent manner. We pharmacologically inhibited gamma-secretase activity using Compound E which specifically blocks SEL-12 but not HOP-1 (another *C. elegans* presenilin homolog) mediated gamma-secretase activity (*Francis et al., 2002*; *Sarasija and Norman, 2015*). In order to obtain complete lack of gamma-secretase activity, we treated *hop-1(ar179)* null animals with Compound E and examined their response to light touch. The gamma-secretase inhibited animals responded at

87.1, 79.8 and 78.3% to light touch at day 1, 2 and 3 of adulthood respectively, which is similar to wild type animals at 83.1, 69.3 and 67.3% (*Figure 4B*). Furthermore, analysis of ALM mechanosensory neuron morphology in animals treated with Compound E did not show ectopic sprouts as is observed in *sel-12* mutants (*Figure 4C*). While compound E treatment clearly shows loss of gamma-secretase function regarding Notch signaling (e.g. *glp-1* like sterility), it is unclear whether the drug is targeting the nervous system. To address this caveat, we generated a proteolytic dead version of SEL-12. The gamma-secretase activity of presenilin is dependent on the presence of two intact aspartate residues, D257 and D385 in PSEN1. Disruption of one aspartate residue leaves PS catalytically nonfunctional (*Kim et al., 2005*; *Wolfe et al., 1999*). Thus, to disrupt aspartyl proteolytic activity of SEL-12, we generated a D226A mutation (equivalent to D257A in PSEN1) in our *sel-12* genomic rescue construct and tested whether this mutant construct could rescue the light touch defect observed in *sel-12* mutants. Consistent with gamma-secretase inactivation, the SEL-12 D226A mutant does not rescue the loss of Notch signaling required for vulval development leaving the *sel-12* animals with their characteristic egg-laying defect. However, *sel-12(ty11)* animals expressing the gamma-secretase dead SEL-12, similar to the animals expressing wild type SEL-12 rescue construct show improved response to touch at 85 and 82.8%, respectively, comparable to the response of wild type animals at 79.8% (*Figure 4D*). Similar rescue of mechanosensation was observed in *sel-12 (ar131)* expressing either wild type or gamma-secretase dead SEL-12 (*Figure 3—figure supplement 1B*). In addition, *sel-12* animals expressing the gamma-secretase dead SEL-12 rescued the neurodegenerative phenotypes, including waves, lesions, breaks, and ectopic sprouting, comparable to the animals expressing wild type SEL-12 (*Figure 4—figure supplement 1*). These data indicate that gamma-secretase activity does not have a role in maintenance of mechanosensory neuronal integrity or function.

## Expression of human Abeta1-42 peptide results in loss of mechano-perception in a mechanism disparate from loss of SEL-12

The major histopathological hallmark of Alzheimer's disease is the presence of amyloid plaques; aggregates of Abeta peptides formed from the consecutive cleavage of APP by beta and gamma secretases. Since *C. elegans* do not form Abeta peptides due to their lack of beta-secretase and APP (*Daigle and Li, 1993*; *McColl et al., 2012*), we used a transgenic line, which overexpresses human Abeta1-42 peptide in the nervous system (pan-neuronal expression) to investigate whether Abeta overexpression causes similar phenotypes as *sel-12* mutants (*Wu et al., 2006*). Previous analyses of these Abeta1-42 overexpressing animals have demonstrated neuronal dysfunction (*Ahmad and Ebert, 2017*; *Bravo et al., 2018*; *Dosanjh et al., 2010*; *Wu et al., 2006*). Thus, it is clear that Abeta1-42 overexpression is toxic and causes cellular defects. Here, we find that pan-neuronal Abeta1-42 expressing day one adult animals (*Figure 5—figure supplement 1*) show a mechanosensory deficit similar to that of *sel-12* mutants (*Figure 5A*); however, they do not show any neuronal structural aberrations (*Figure 5B,C*) as we observed in *sel-12* mutants. Furthermore, both mitochondrial structure and mitochondrial calcium levels in the Abeta1-42 overexpressing animals appear wild type (*Figure 5D,E*). These data suggest that while the ectopic overexpression of Abeta1-42 likely perturbs neuronal function, unlike *sel-12* mutants, it appears to be doing so without affecting axon morphology and mitochondrial calcium homeostasis.

## Decreasing ER Ca$^{2+}$ release improves mitochondrial function and suppresses neurodegeneration in *sel-12* mutants

Since we previously demonstrated that reducing ER Ca$^{2+}$ release in *sel-12* mutants markedly improved the mitochondrial disorganization that occurs in body wall muscle (*Sarasija and Norman, 2015*) and given the presence of elevated mitochondrial Ca$^{2+}$ in *sel-12* mutants (*Figure 1A*, *Figure 1—figure supplement 1B*), we hypothesized that exacerbated ER Ca$^{2+}$ signaling in the nervous system is promoting the mechanosensory neuronal defects observed in these animals by causing heightened OXPHOS mediated ROS production. To test this hypothesis, we introduced a calreticulin null mutation, *crt-1(jh101)*, which reduces ER Ca$^{2+}$ release (*Michalak et al., 1999*; *Xu et al., 2001*), into the *sel-12* mutant background, and determined whether the *crt-1* null mutation could alter mitochondrial Ca$^{2+}$ load using a mitochondrial targeted GCaMP6/mCherry Ca$^{2+}$ sensor. From these analyses, we found that *crt-1* null animals have a significantly reduced GCaMP6

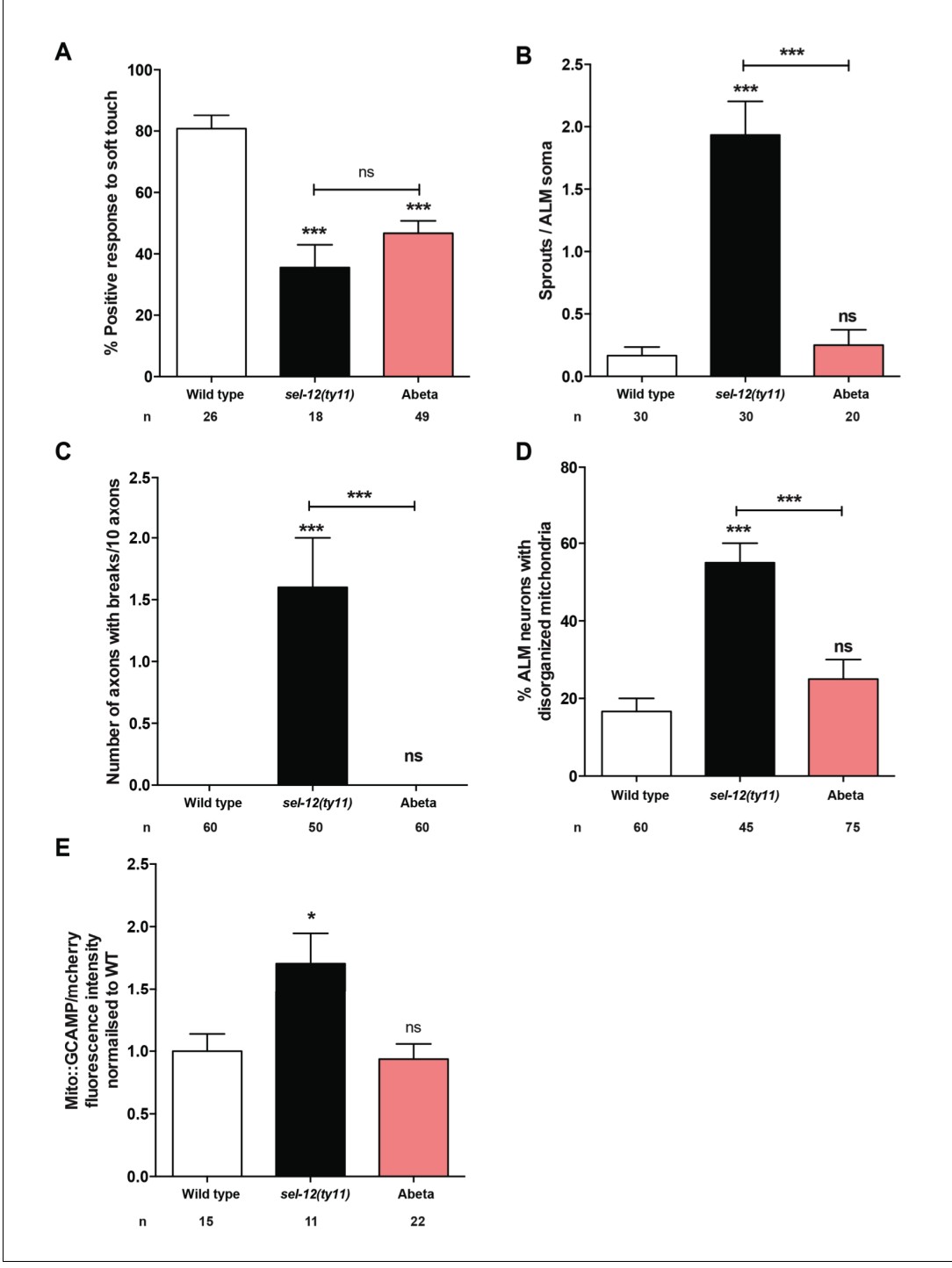

**Figure 5.** Expression of human Abeta1-42 peptide in the the nervous system results in functional loss dissimilar from *sel-12* mutants. (**A**) Quantification of the response of Abeta peptide expressing animals to light anterior and posterior touch at day 1 of adulthood. (**B**) Quantification of aberrant structures (sprouts/branches) on ALM neuronal soma and (**C**) frequency of breaks in ALM and PLM neuronal processes at day 1 of adulthood. (**D**) Quantification of the incidence of discontinuous ALM neuronal mitochondria. Analysis was done using *twnEx8* transgenic animals. (**E**) Quantification of mitochondrial Ca$^{2+}$ using animals expressing mito::GCaMP6 and mCherry in their mechanosensory neurons (*takEx415*). n = number of animals analyzed per genotype. Data are displayed as mean ± SEM, and all comparisons have been made to wild type animals unless otherwise indicated. ns $p > 0.05$, *$p < 0.05$, ***$p < 0.0001$ (One way ANOVA with Tukey test).

DOI: https://doi.org/10.7554/eLife.33052.013

*Figure 5 continued on next page*

*Figure 5 continued*

The following source data and figure supplements are available for figure 5:

**Source data 1.** Raw data for *Figure 5*.
DOI: https://doi.org/10.7554/eLife.33052.015
**Figure supplement 1.** Western analysis of pan-neuronal expression of Abeta1-42 in *C. elegans*.
DOI: https://doi.org/10.7554/eLife.33052.014
**Figure supplement 1—source data 1.** Raw data for *Figure 5—figure supplement 1*.
DOI: https://doi.org/10.7554/eLife.33052.016

fluorescence compared to wild type animals and the introduction of the *crt-1* null mutation into *sel-12(ty11)* animals normalized their GCaMP6 fluorescence to wild type levels (*Figure 6A*). Similarly, we found that introduction of an *unc-68* null mutation (*unc-68* encodes the only ryanodine receptor in the *C. elegans* genome) into the *sel-12* mutant background also normalized the mitochondrial GCaMP6 fluorescence to wild type levels (*Figure 6—figure supplement 1A*).

Next, we explored the impact this normalization of mitochondrial $Ca^{2+}$ level has on mitochondrial disorganization, OCR and ROS levels in *sel-12* mutants. Similar to our studies of mitochondrial structure in the body wall muscle (*Sarasija and Norman, 2015*), we observe a dramatic improvement of mitochondrial organization in *crt-1; sel-12* double mutants compared to *sel-12* mutants (*Figure 6B, C*). Next, to ascertain whether exacerbated ER $Ca^{2+}$ release is responsible for the elevated OCR observed in *sel-12* mutants, we determined the OCR in *crt-1; sel-12* and *unc-68; sel-12* mutant animals. We found that *crt-1; sel-12* and *unc-68; sel-12* double mutants have basal OCR and maximal OCR rates comparable to wild type and significantly lower than *sel-12* (*Figure 6D,E*, *Figure 6—figure supplement 1B,C*). These data are consistent with the notion that higher $Ca^{2+}$ release from the ER leads to elevated OXPHOS in *sel-12* mutants. Additionally, this elevated OXPHOS activity can explain the increased levels of ROS we previously observed in *sel-12* mutants (*Sarasija and Norman, 2015*). Since we found that the OCR was decreased in *sel-12* animals that had reduced ER $Ca^{2+}$ release, indicating decreased OXPHOS activity, we investigated whether *crt-1; sel-12* mutant animals would have a reduced ROS burden. To test this, we quantified ROS formation and discovered that *crt-1; sel-12* animals have ROS levels comparable to wild type animals and lower than *sel-12* mutants animals (*Figure 6F*). These data suggest that mutations in *sel-12* lead to exacerbated ER $Ca^{2+}$ release, which causes mitochondrial morphology changes and increased metabolic activity resulting in elevated ROS.

In order to determine whether the improvement of mitochondrial organization and function of *sel-12* mutants in the presence of the *crt-1* or *unc-68* null mutation could rescue the *sel-12* neurodegenerative phenotypes, we assessed mechanosensory neuron morphology in *crt-1; sel-12* and *unc-68; sel-12* double mutants and behavioral response to light touch in *crt-1; sel-12* double mutants. Strikingly, the ectopic sprouting, wave-like processes, lesions and breaks of the ALM mechanosensory neurons observed in *sel-12* mutants are strongly suppressed in *crt-1; sel-12* and *unc-68; sel-12* animals (*Figure 6G*, *Figure 6—figure supplement 1D–E*, *Figure 6—figure supplement 2A–C*). Consistent with the absence of neurodegenerative morphology of the mechanosensory neurons, the *crt-1; sel-12* animals also showed improved response to light touch with a 77.7% response compared to 48.4% response of *sel-12* animals (*Figure 6H*). This suggests that loss of SEL-12 function results in deregulated ER $Ca^{2+}$ release which promotes neuronal mitochondrial disorganization, enhanced OXPHOS and high ROS production which leads to neurodegeneration in *sel-12* mutants.

## Reducing mitochondrial $Ca^{2+}$ uptake improves mitochondrial function and suppresses neurodegeneration in *sel-12* mutants

$Ca^{2+}$ uptake into the mitochondrial matrix is mediated by the highly conserved mitochondrial $Ca^{2+}$ uniporter (*Baughman et al., 2011*; *Csordás et al., 2013*; *De Stefani et al., 2011*). Prior work from our lab has shown that the introduction of a null mutation in the gene encoding the mitochondrial $Ca^{2+}$ uniporter, *mcu-1*, which reduces mitochondrial $Ca^{2+}$ uptake (*Xu and Chisholm, 2014*), into *sel-12(ty11)* animals improves the body wall muscle mitochondria structure similar to the *crt-1* mutation in *sel-12(ty11)* animals (*Sarasija and Norman, 2015*). Therefore, we investigated the role mitochondrial $Ca^{2+}$ uptake has on the neurodegenerative phenotypes observed in *sel-12(ty11)* animals by

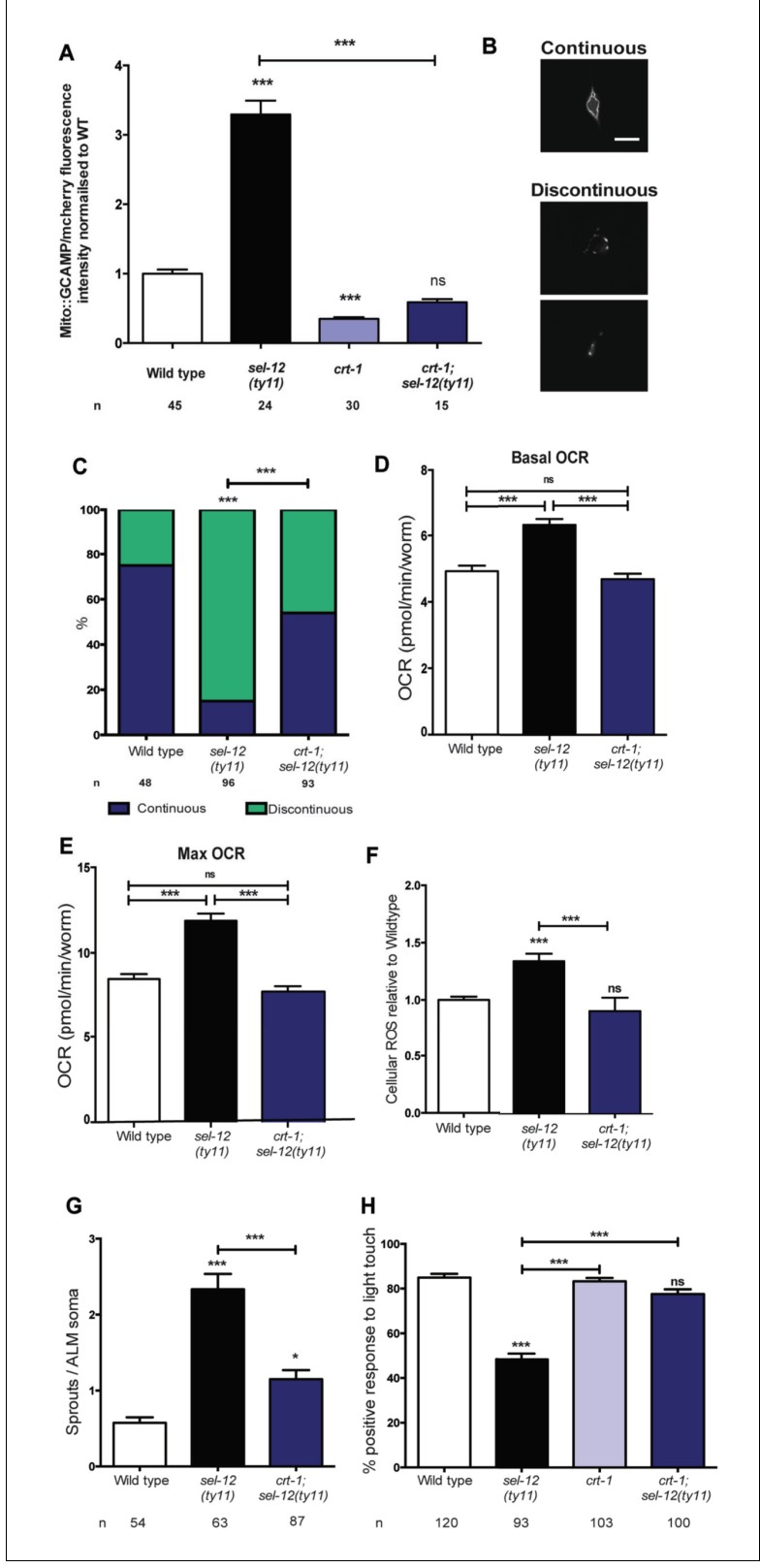

**Figure 6.** Reducing ER Ca²⁺ release in *sel-12* mutants rescues mitochondrial disorganization and dysfunction, ectopic neurite sprouting and mechanosensory defects. (**A**) Quantification of mitochondrial Ca²⁺ using animals expressing mito::GCaMP6 and mCherry in their body wall muscle (*takEx347*). n = 30 per strain. (**B**) Representative images and (**C**) quantification of ALM neuronal mitochondrial morphology. Scale bar represents 10 μm. Analysis
*Figure 6 continued on next page*

*Figure 6 continued*

was done using transgenic animals expressing mito::GFP in mechanosensory neurons (*jsIs609*). Data are displayed with blue and green representing the percentage of animals displaying continuous and discontinuous mitochondrial morphology, respectively. ***p<0.001 (Chi-square test). (D) Basal and (E) maximal respiration rates (after exposure to FCCP) of wild type, *sel-12(ty11)*, and *crt-1;sel-12(ty11)* animals. Data are from three replicate assays. (F) $H_2DCF$-DA assay measuring ROS levels relative to wild type. Data are from three replicate assays. (G) Quantification of aberrant neuronal structures (sprouts/branches) in ALM neurons in day one animals. ALM neuronal morphology analysis is done using transgenic animals expressing *mec-4p*::GFP (*zdIs5*). (H) Quantification of the response of day one adult animals to light anterior and posterior touch. n = number of animals analyzed per genotype. Data are displayed as mean ± SEM, and all comparisons have been made to wild type animals unless otherwise indicated. ns p>0.05, *p<0.05, ***p<0.0001 (One way ANOVA with Tukey test).
DOI: https://doi.org/10.7554/eLife.33052.021

The following source data and figure supplements are available for figure 6:

**Source data 1.** Raw data for *Figure 6*.
DOI: https://doi.org/10.7554/eLife.33052.024
**Figure supplement 1.** Loss of ryanodine receptors results in the improvement of neurodegeneration in *sel-12* null animals.
DOI: https://doi.org/10.7554/eLife.33052.022
**Figure supplement 1—source data 1.** Raw data for *Figure 6—figure supplement 1*.
DOI: https://doi.org/10.7554/eLife.33052.025
**Figure supplement 2.** Reduction of ER $Ca^{2+}$ release improves the structure of mechanosensory neurons in *sel-12* animals.
DOI: https://doi.org/10.7554/eLife.33052.023
**Figure supplement 2—source data 2.** Raw data for *Figure 6—figure supplement 2*.
DOI: https://doi.org/10.7554/eLife.33052.026

generating *mcu-1; sel-12(ty11)* double mutants. In order to determine the effect the absence of the mitochondrial $Ca^{2+}$ uniporter has on mitochondrial $Ca^{2+}$ levels, we measured mitochondrial $Ca^{2+}$ levels and found that *mcu-1* null animals have reduced mitochondrial $Ca^{2+}$ levels compared to wild type animals (*Figure 7A*). Also, akin to our analyses of the *crt-1; sel-12* and *unc-68; sel-12* double mutants, *mcu-1; sel-12(ty11)* animals, show a reduction of mitochondrial $Ca^{2+}$ load (*Figure 7A*). Furthermore, as we previously observed in the body wall muscle (*Sarasija and Norman, 2015*), we found that the *mcu-1* mutation improved the organization of neuronal mitochondria in *sel-12* mutants (*Figure 7B*).

Consistent with the improvement in mitochondrial structure and $Ca^{2+}$ load, *mcu-1; sel-12(ty11)* animals also show normal ROS and OCR levels similar to what was observed in *sel-12* mutants with reduced ER $Ca^{2+}$ release (*Figure 7C–E*). Along with these observations, phenotypes of structural neurodegeneration such as ectopic sprouting, wave-like processes, lesions and breaks are also decreased in the *mcu-1; sel-12(ty11)* animals compared with *sel-12(ty11)* mutants (*Figure 7F*, *Figure 7—figure supplement 1A–C*). Moreover, the *mcu-1; sel-12(ty11)* animals show 78.8% response to light touch compared to 48.4% response of *sel-12(ty11)* animals (*Figure 7G*). Taken together these data suggest that reducing mitochondrial $Ca^{2+}$ uptake normalizes mitochondrial metabolic activity and suppresses the neurodegenerative phenotype observed in *sel-12(ty11)* animals.

Next, to determine whether reducing mitochondrial $Ca^{2+}$ uptake could alleviate the mitochondrial metabolic dysregulation observed in FAD fibroblasts (*Figure 1E–H*), we treated control fibroblasts and fibroblasts isolated from FAD patients with Ruthenium360 (Ru360). Ru360 is a cell-permeable mitochondrial $Ca^{2+}$ uniporter inhibitor. Consistent with the improvement in mitochondrial function observed in the *mcu-1; sel-12(ty11)* animals, treatment of FAD fibroblasts with Ru360 resulted in a decrease of both ATP and ROS levels in these cells (*Figure 7H,I*).

## Reducing mitochondrial function but not mitochondrial fission rescues neurodegeneration in *sel-12* mutants

Previously, we found that reducing mitochondrial fission in *sel-12* mutants could improve mitochondrial morphology (*Sarasija and Norman, 2015*). This was accomplished by knocking down the gene encoding DRP-1 by RNA interference (RNAi). DRP-1 is a dynamin-1-like protein that regulates

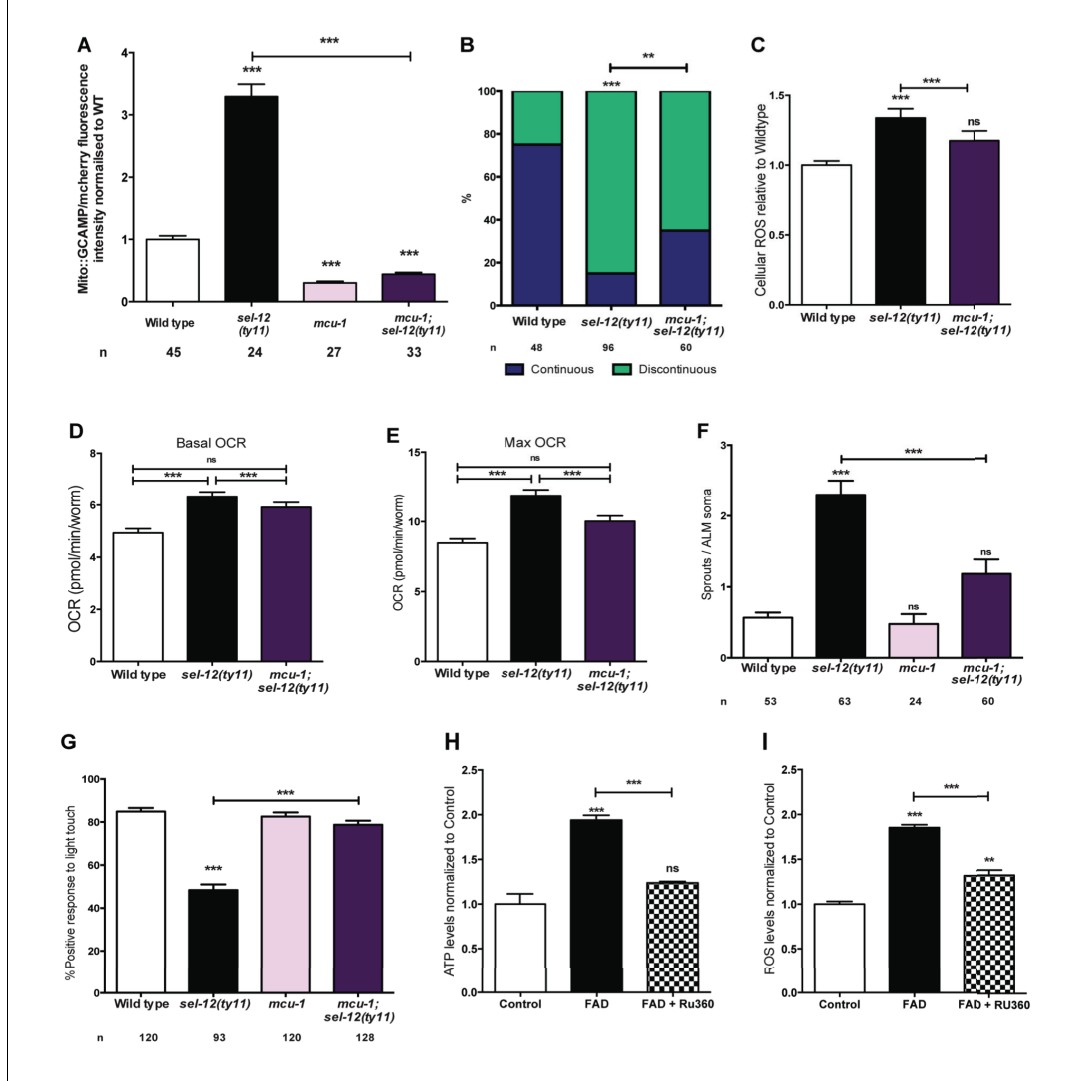

**Figure 7.** PSEN functions to regulate mitochondrial activity by mediating Ca$^{2+}$ transfer from the ER to the mitochondria. (**A**) Quantification of mitochondrial Ca$^{2+}$ using animals expressing mito::GCaMP6 and mCherry in their body wall muscle (*takEx347*). (**B**) Quantification of ALM neuronal mitochondrial morphology. Analysis was done using transgenic animals expressing mito::GFP in mechanosensory neurons (*jsIs609*). Data are displayed with blue and green representing the percentage of animals displaying continuous and discontinuous mitochondrial morphology, respectively. **p<0.01, ***p<0.001 (Chi-square test). (**C**) H$_2$DCF-DA assay measuring ROS levels relative to wild type. Data are from three replicate assays. (**D**) Basal and (**E**) maximal respiration rates (after exposure to FCCP) of wild type, *sel-12(ty11)*, and *mcu-1;sel-12(ty11)* animals. Data are from three replicate assays. (**F**) Quantification of aberrant neuronal structures (sprouts/branches) in ALM neurons in day one animals. ALM neuronal morphology analysis is done using transgenic animals expressing *mec-4p*::GFP (*zdIs5*). (**G**) Quantification of the response of day one wild type, *sel-12(ty11)*, *mcu-1;sel-12(ty11)* and *mcu-1* animals to light anterior and posterior touch. (**H**) ATP and (**I**) ROS levels normalized to protein content in skin fibroblasts isolated from control and FAD patients, treated with Ru360. n = number of animals analyzed per genotype. Data are displayed as mean ± SEM, and all comparisons have been made to wild type animals unless otherwise indicated). ns p>0.05, ***p<0.0001 (One way ANOVA with Tukey test).
DOI: https://doi.org/10.7554/eLife.33052.027

The following source data and figure supplements are available for figure 7:

**Source data 1.** Raw data for *Figure 7*.
DOI: https://doi.org/10.7554/eLife.33052.029

**Figure supplement 1.** Reduction of ER Ca$^{2+}$ release improves the structure of mechanosensory neurons in *sel-12* animals.
DOI: https://doi.org/10.7554/eLife.33052.028

**Figure supplement 1—source data 1.** Raw data for *Figure 7—figure supplement 1*.
DOI: https://doi.org/10.7554/eLife.33052.030

mitochondrial fission (*Labrousse et al., 1999*). Thus, we sought to determine whether inhibiting mitochondrial fission could normalize mitochondrial activity of the *sel-12* mutants and prevent neurodegeneration. We knocked down the expression of *drp-1* in *sel-12* mutants by *drp-1(RNAi)* and analyzed mitochondrial structure, mitochondrial calcium levels, OXPHOS status, neuronal structure, and response to soft touch in these animals. While *drp-1(RNAi)* rescues the mitochondrial structure in the *sel-12(ty11)* animals (*Figure 8—figure supplement 1A*), it is unable to restore mitochondrial calcium levels, OCR, and neurodegeneration (*Figure 8—figure supplement 1B–F*). This data suggests that defects in mitochondrial morphology may be a symptom but not the cause of the neurodegeneration observed in *sel-12* mutants.

Since we have observed high mitochondrial activity in young adult *sel-12* mutants and that the reduction of ER $Ca^{2+}$ release or mitochondrial $Ca^{2+}$ uptake can reduce mitochondrial activity and improve the neurodegeneration observed in *sel-12* mutants, we next sought to determine whether directly reducing mitochondrial activity could suppress the *sel-12* neurodegenerative phenotype. To reduce mitochondrial activity, we used doxycycline to inhibit mitochondrial function (*Moullan et al., 2015*). *sel-12(ty11)* animals grown in the presence of doxycycline showed inhibition of mitochondrial function as evidenced by the suppression of the high mitochondrial respiration rate observed in *sel-12* mutants (*Figure 8A,B*). Interestingly, doxycycline treatment also rescued the mechanosensory defects in *sel-12(ty11)* animals, both in touch response and axon morphology (*Figure 8C–E*). Moreover, to test whether elevated mitochondrial activity causes the elevated ATP and ROS levels and OCR observed in FAD fibroblasts, we treated control and FAD fibroblasts with doxycycline and measured these outputs. Consistent with FAD cells showing high mitochondrial activity, doxycycline treatment reduced ATP and ROS levels as well as OCR in FAD fibroblasts (*Figure 8F–I*). These data taken together with our observations of increased mitochondrial $Ca^{2+}$ levels in *sel-12* mutants and the ability of reducing $Ca^{2+}$ transfer from the ER to the mitochondrial in suppressing the neurodegeneration associated with *sel-12* mutants indicates that exacerbated ER $Ca^{2+}$ transfer to the mitochondria boosts mitochondrial activity causing neurodegeneration in the *sel-12* mutants.

## Mitochondrial generated superoxides cause mechanosensation and neuronal morphology defects in *sel-12* mutants

Thus far, our analyses have found that mutations in *sel-12* result in deregulation of ER $Ca^{2+}$ release and mitochondrial $Ca^{2+}$ uptake, which leads to increased OXPHOS, and increased global ATP as well as ROS levels and results in neurodegeneration. To gain further insight into the cause of neurodegeneration in *sel-12* mutants, we sought to determine the mechanism underlying the neurodegenerative phenotypes observed in *sel-12* mutants. First, to investigate whether there is an elevation in mitochondrial ROS in the mechanosensory neurons undergoing neurodegeneration, we used a mitochondrial matrix targeted roGFP, a reduction-oxidation sensitive green fluorescent protein (*Cannon and Remington, 2008*; *Melentijevic et al., 2017*) and discovered that *sel-12(ty11)* animals have elevated mitochondrial ROS compared to wild type animals (*Figure 9A*). Since OXPHOS leads to the production of superoxides, high levels of which can be toxic to cells and we observed elevated OXPHOS and mitochondrial ROS in *sel-12* mutants, we hypothesized that mitochondrial superoxide production could be causing the neurodegeneration observed in *sel-12* mutants. To investigate the impact of mitochondrial superoxide production on the mechanosensory defects observed in *sel-12* mutants, we exposed *sel-12(ty11)* animals to a mitochondria-targeted superoxide-scavenger called MitoTEMPO or triphenylphosphonium (TPP), which lacks the superoxide-scavenging moiety and acts as a control. Consistent with MitoTEMPO reducing mitochondrial ROS levels without altering mitochondrial $Ca^{2+}$ levels, we found that *sel-12(ty11)* animals treated with 500 µM MitoTEMPO showed significant reduction in oxidized neuronal mitochondrial roGFP fluorescence levels (*Figure 9—figure supplement 1A*) while mitochondrial $Ca^{2+}$ levels remained similar to control *sel-12(ty11)* animals (*Figure 9—figure supplement 1B*). Next, we examined neuronal structure and behavioral response to light touch. *sel-12(ty11)* animals raised on MitoTEMPO show a decrease in the number of ectopic neuronal sprouts from two sprouts/ALM in control *sel-12(ty11)* animals to one sprouts/ALM (*Figure 9B*) while TPP treated animals showed no such improvement (*Figure 9B*). Also, MitoTEMPO treatment resulted in a normal behavioral response to light touch at 83.1% similar to wild type animals at 82.6% and much improved compared to the control and TPP treated *sel-12(ty11)* animals (*Figure 9C*). However, MitoTEMPO treatment does not rescue the mitochondrial morphology defects seen in *sel-12(ty11)* animals (*Figure 9—figure supplement 1C*), suggesting that

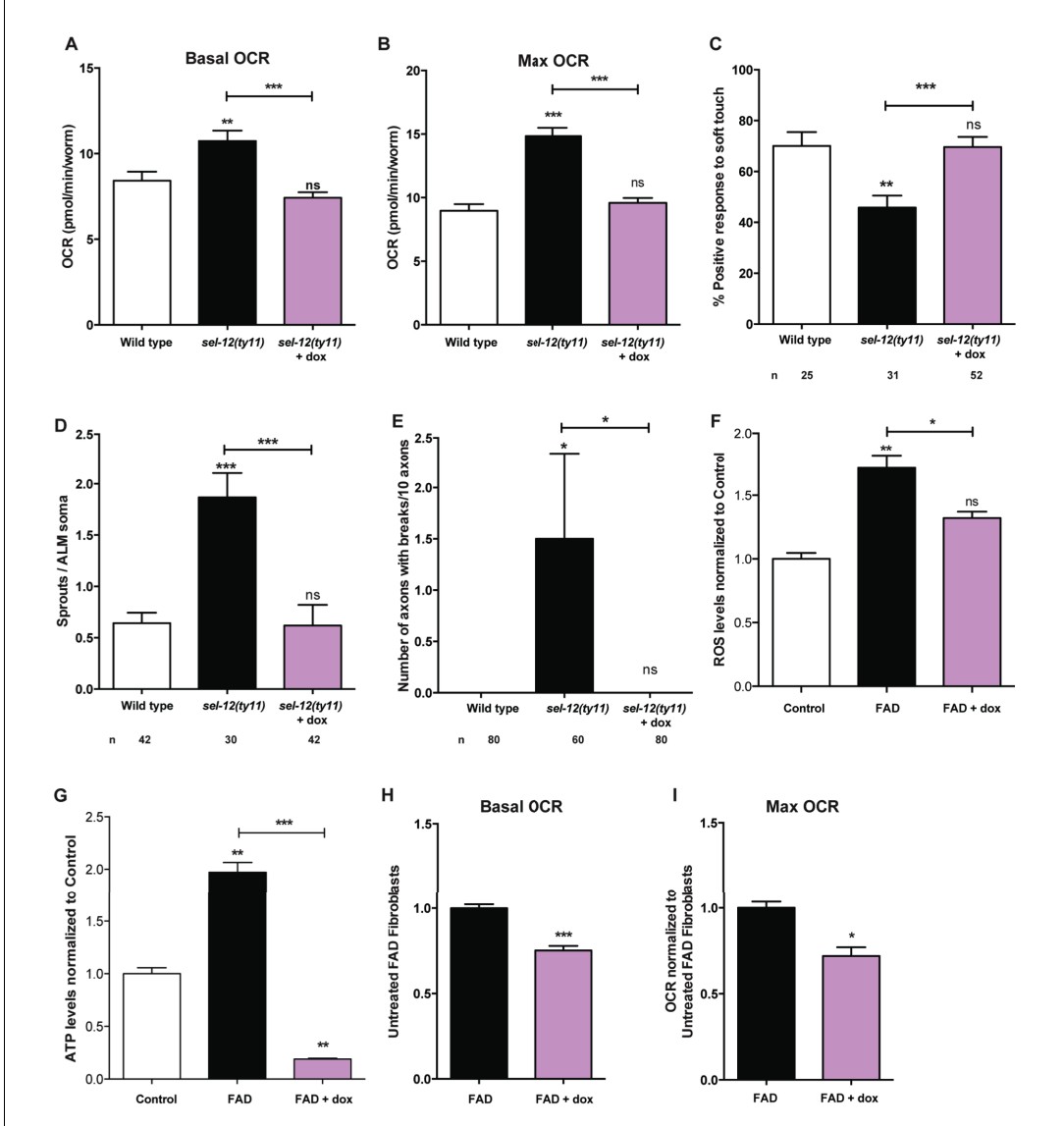

**Figure 8.** Reducing mitochondrial activity using doxycyline alleviates neurodegenerative phenotypes observed in *sel-12* mutants. (**A**) Basal and (**B**) maximal respiration rates of wild type and *sel-12* mutant animals treated with doxycycline (dox). (**C**) Quantification of the response of dox treated day one *sel-12(ty11)* animals to light anterior and posterior touch. (**D**) Quantification of aberrant structures (sprouts/branches) on ALM neuronal soma and (**E**) frequency of breaks in ALM and PLM neuronal processes at day 1 of adulthood in response to dox. (**F**) ROS and (**G**) ATP levels normalized to protein content in skin fibroblasts isolated from control and FAD patients treated with dox. (**H**) Basal and (**B**) maximal respiration rates in skin fibroblasts isolated from control and FAD patients treated with dox. n = number of animals analyzed per genotype. Data are displayed as mean ± SEM, and all comparisons have been made to wild type animals unless otherwise indicated. ns p>0.05, *p<0.05, **p<0.001, ***p<0.0001 (One way ANOVA with Tukey test for A-G, two-tailed T-test for H and I).
DOI: https://doi.org/10.7554/eLife.33052.031

The following source data and figure supplements are available for figure 8:

**Source data 1.** Raw data for *Figure 8.*
DOI: https://doi.org/10.7554/eLife.33052.033

**Figure supplement 1.** Reduction of mitochondrial fission does not alleviate the neurodegeneration in *sel-12(ty11)* animals.
DOI: https://doi.org/10.7554/eLife.33052.032

**Figure supplement 1—source data 1.** Raw data for *Figure 8—figure supplement 1*.
DOI: https://doi.org/10.7554/eLife.33052.034

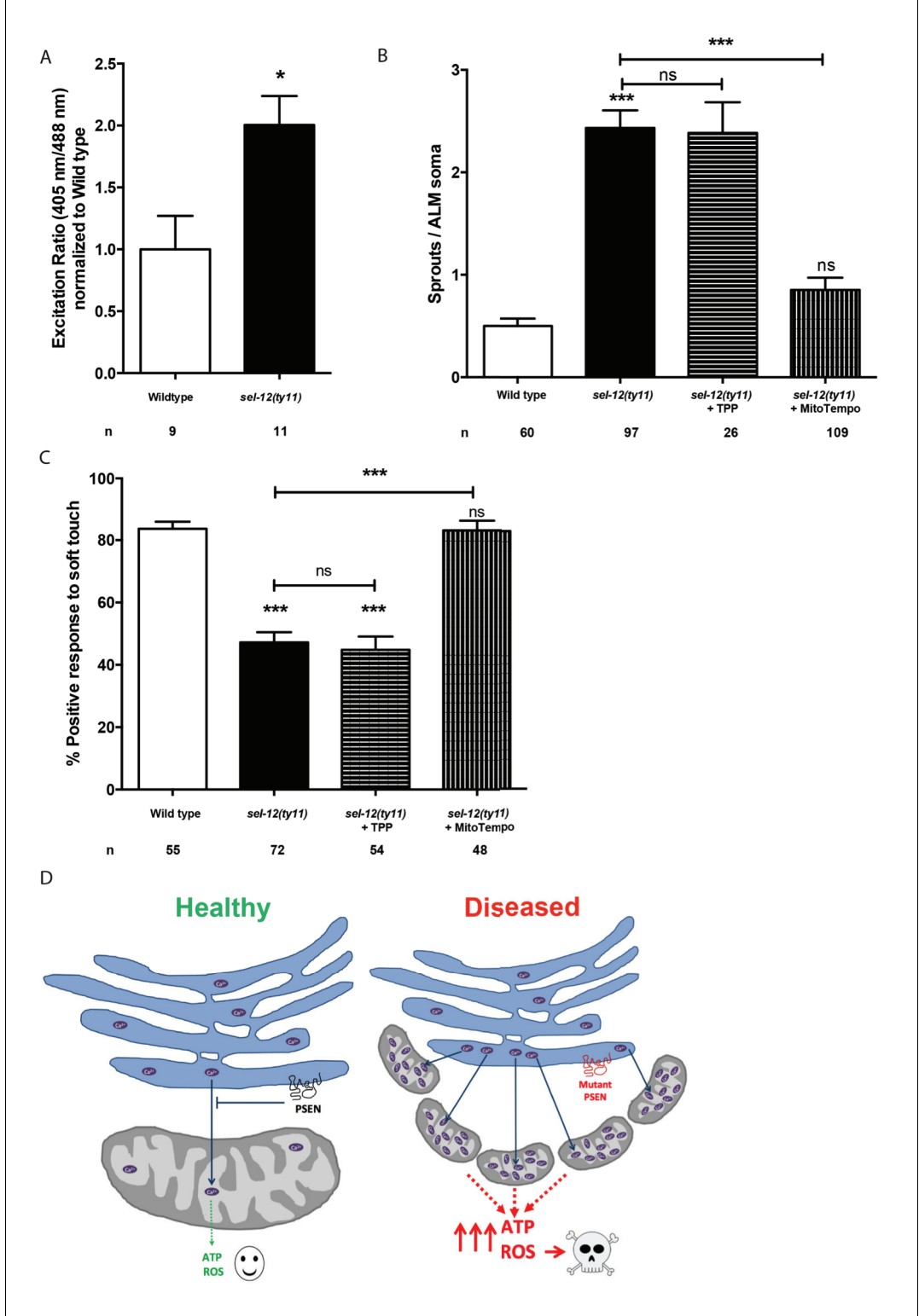

**Figure 9.** MitoTEMPO supplementation improves neuronal structure and function of *sel-12* mutants. (**A**) Quantification of oxidized state of ALM neuronal mitochondria using *zhsEx17* [*Pmec-4*mitoLS::ROGFP] (**B**) Quantification of aberrant neuronal structures (sprouts/branches) in ALM neurons in day one animals. ALM neuronal morphology analysis is done using transgenic animals expressing *mec-4p*::GFP (*zdIs5*). (**C**) Quantification of the response of day one animals to light anterior and posterior touch. n = number of animals analyzed per genotype. Data are displayed as mean ± SEM, and all comparisons have been made to wild type animals unless

*Figure 9 continued on next page*

*Figure 9 continued*

otherwise indicated). ns p>0.05, *p<0.05, ***p<0.0001 (One way ANOVA with Tukey test and two-tailed t-test for A). (**D**) Model: Wild type presenilin regulates the normal transfer of $Ca^{2+}$ (purple ovals) from the ER to the mitochondria. When presenilin is mutated or absent, excessive ER-mitochondria $Ca^{2+}$ transfer results in mitochondrial morphological changes, increased mitochondrial respiration and ROS production causing neurodegeneration.

DOI: https://doi.org/10.7554/eLife.33052.035

The following source data and figure supplements are available for figure 9:

**Source data 1.** Raw data for *Figure 9*.
DOI: https://doi.org/10.7554/eLife.33052.037

**Figure supplement 1.** Mitochondrial disorganization in the mechanosensory neurons of *sel-12* animals is not a direct result of elevated ROS levels.
DOI: https://doi.org/10.7554/eLife.33052.036

**Figure supplement 1—source data 1.** Raw data for *Figure 9—figure supplement 1*.
DOI: https://doi.org/10.7554/eLife.33052.038

oxidative stress is not directly causing mitochondrial structural disorganization. Together our data suggest that deregulated $Ca^{2+}$ transfer from the ER to the mitochondria increases OXPHOS resulting in superoxide generation triggering oxidative stress mediated neurodegeneration.

## Discussion

Mutations in the genes encoding *PSEN1* and *PSEN2* are the most frequent cause of FAD. Despite the identification of PSEN involvement in AD over 20 years ago, the mechanism causing neurodegeneration in FAD patients has remained elusive. In this study, we provide evidence that mutations in the gene encoding a *C. elegans* PSEN homolog, *sel-12* result in ER to mitochondria $Ca^{2+}$ signaling defects, elevated mitochondrial $Ca^{2+}$ levels, mitochondrial structural disorganization, dysfunction of mitochondrial metabolism and subsequent oxidative stress mediated neurodegeneration. We also demonstrate that the structural and functional neurodegeneration observed in *sel-12* mutants are independent of the gamma-secretase activity of SEL-12 and are a result of mitochondria generated superoxide that arise due to increased ER to mitochondria $Ca^{2+}$ signaling resulting in amplified OXPHOS. Moreover, we show that skin fibroblasts isolated from FAD patients with mutations in *PSEN1* show a similar mitochondrial phenotype as observed in *C. elegans sel-12* mutants, demonstrating a conserved role of PSEN in mitochondrial regulation. While dysregulation in ER $Ca^{2+}$ signaling (*Chan et al., 2000*; *Cheung et al., 2008*; *Green et al., 2008*; *Leissring et al., 1999*; *Stutzmann et al., 2004*), mitochondrial dysfunction (*Area-Gomez et al., 2012*; *Kipanyula et al., 2012*; *Zampese et al., 2011*), and deleterious effects of ROS have been observed separately in various in-vivo and in-vitro models of AD, here we provide evidence that link these observations and demonstrate the central role played by mitochondrial metabolism in AD pathogenesis using an intact animal model of AD where neurodegeneration arises due to altered ER to mitochondrial $Ca^{2+}$ signaling, in the absence of Abeta peptides.

In order to maintain neuronal function and health, optimal mitochondrial activity is critical to provide energy and concomitantly manage ROS production. Furthermore, mitochondria have a critical role in buffering $Ca^{2+}$ and $Ca^{2+}$ in turn modulates mitochondrial activity. Indeed, increased mitochondrial $Ca^{2+}$ aids in the allosteric activation of several TCA cycle enzymes, including pyruvate dehydrogenase α-ketoglutarate dehydrogenase, and isocitrate dehydrogenase (*McCormack and Denton, 1993*), stimulates the ATP synthase (complex V) (*Das and Harris, 1990*), α-glycerophosphate dehydrogenase (*McCormack and Denton, 1993*; *Wernette et al., 1981*) and the adenine nucleotide translocase (ANT) (*McCormack and Denton, 1993*; *Mildaziene et al., 1995*). Thus, the increase in mitochondrial $Ca^{2+}$ concentration results in the coordinated upregulation of the TCA cycle and OXPHOS machinery, allowing for increased oxygen consumption, and ATP and ROS production. Here, we discover that mutations in *sel-12* lead to deregulated ER and mitochondrial $Ca^{2+}$ signaling resulting in increased mitochondrial $Ca^{2+}$ levels, which promotes mitochondrial metabolic activity. Additionally, we provide evidence that this elevated metabolic activity elevates ROS levels

in *sel-12* mutants and the deleterious effects of ROS results in the emergence of structural and functional phenotypes of neurodegeneration.

Upon reducing ER $Ca^{2+}$ release or mitochondrial $Ca^{2+}$ uptake, the mitochondria of *sel-12* mutants are no longer disorganized and show normal OCR, and the high levels of ROS and neurodegenerative phenotypes observed in *sel-12* mutants are suppressed. These data indicate that SEL-12 activity is required for normal ER to mitochondrial $Ca^{2+}$ signaling, proper mitochondrial function and mitochondrial superoxide maintenance. Consistent with PSENs having a role in ER $Ca^{2+}$ regulation, several studies in a variety of PSEN FAD models as well as tissue samples isolated from AD patients have shown that loss of PSEN function leads to perturbed ER $Ca^{2+}$ signaling (*Chan et al., 2000*; *Green et al., 2008*; *Ito et al., 1994*; *Leissring et al., 1999*; *Smith et al., 2005*; *Tu et al., 2006*). Furthermore, PSENs are predominantly ER membrane proteins that are enriched in the mitochondria-associated ER membrane (*Area-Gomez et al., 2009*). The mitochondria-associated ER membrane is a highly active subcompartment of the ER that is physically connected to the mitochondria and is critical for several metabolic functions, such as $Ca^{2+}$ homeostasis, cholesterol metabolism and the synthesis and transfer of phospholipids between the ER and mitochondria. Importantly, elevated ER to mitochondrial contact and crosstalk has been observed in *PSEN1* and *PSEN2* knockout cells, in cells expressing FAD mutant PSEN2, and in skin fibroblasts from FAD and sporadic AD patients (*Area-Gomez et al., 2012*; *Filadi et al., 2016*; *Kipanyula et al., 2012*; *Zampese et al., 2011*). Although the functional importance of these observations has not been resolved, our analyses of skin fibroblasts isolated from FAD patients, indicates that there is an up regulation of OXPHOS, ATP production and ROS generation that can be reduced by inhibiting mitochondrial $Ca^{2+}$ uptake, similar to what we observe in *sel-12* mutants. We also demonstrate that in *sel-12* mutants neurodegeneration is caused by elevated ER to mitochondrial transfer of $Ca^{2+}$. Taken together, these data present an inclusive model that indicates that PSEN function on the ER membrane is required for normal ER $Ca^{2+}$ transfer to the mitochondria and in the absence of optimal PSEN function, excessive $Ca^{2+}$ is transferred to the mitochondria leading to mitochondrial dysfunction and an elevation of mitochondrial generated superoxide radicals that promote neurodegeneration (*Figure 9D*). However, recent analyses of PSEN1/2 knockout mouse embryonic fibroblasts (MEFs), unlike our studies, have shown reduced mitochondrial activity suggesting PSEN function is vital for mitochondrial function (*Contino et al., 2017*; *Pera et al., 2017*). Significantly, one of these studies demonstrated that the reduction of mitochondrial function in PSEN1/2 KO MEFs could be rescued by PSEN2 and not PSEN1, implicating a role of PSEN2 in mitochondrial function in MEFs (*Contino et al., 2017*). While the reduced mitochondrial activity might appear to be at odds with our study, a recent analysis of astrocytes derived from iPSCs isolated from skin cells from patients with *PSEN1* mutations also showed elevated OCR and ROS production, which could be rescued by CRISPR/Cas9 repair of the *PSEN1* mutation (*Oksanen et al., 2017*). Thus, illustrating an important role of PSEN1 in the regulation of mitochondrial activity in astrocytes similar to SEL-12 in *C. elegans*. Also of note, we did observe in day eight vs. day one adult *sel-12* mutants a drastic reduction of mitochondrial activity in comparison to aged matched wild type animals indicating that in aged animals mitochondria function becomes impaired in *sel-12* mutants, consistent with the decreased mitochondrial function as a result of PSEN mutations (*Figure 1—figure supplement 1C,D*). The impairment of mitochondrial function with age in *sel-12(ty11)* animals could be a result of increased mitochondrial activity resulting in ROS-mediated oxidative damage at earlier stages of adulthood.

The best-studied function of PSEN is its role as the proteolytic component of the gamma-secretase complex and its cleavage of APP and Notch. Gamma-secretase mediated cleavage is required for the activation of Notch signaling and emphasizes the critical role of PSEN in mediating Notch function (*De Strooper et al., 1997*). Also, it is the gamma-secretase mediated cleavage of APP that leads to the generation of Abeta peptides of various lengths, including the toxic Abeta42 (*Bezprozvanny and Mattson, 2008*; *Hardy, 2006*). Here, we provide genetic and pharmacological evidence that gamma-secretase activity of SEL-12 is not required for neurodegenerative phenotypes observed in *sel-12* mutants. While we are not able to rescue Notch signaling defects using a *sel-12* protease dead genetic construct, we are able to rescue the mechanosensory behavioral defects observed in *sel-12* null mutants (*Figure 4D*, *Figure 4—figure supplement 1A-D*). Furthermore, using a gamma-secretase inhibitor, which abolished Notch signaling, we do not recapitulate the neurodegenerative phenotypes observed in *sel-12* mutants (*Figure 4B,C*). Consistently, our analyses of Notch mutants also did not reveal mechanosensory defects (*Figure 4A*). These results demonstrate

that the role of SEL-12 in the regulation of mitochondrial function and nervous system fitness does not require gamma-secretase activity or Notch signaling.

In the last two decades, research into AD therapeutics has relied heavily on the amyloid hypothesis, which holds the toxic Abeta peptides and their aggregation to form plaques responsible for AD pathogenesis. However, the exact role Abeta peptides have in AD is not clear, which is further emphasized by the failure of phase III trials of anti-Abeta therapies like the gamma-secretase inhibitor, semagacestat (*Doody et al., 2013*) and solanezumab, a monoclonal antibody targeting amyloid plaques (*Le Couteur et al., 2016*). This suggests the need to shift focus from Abeta peptides to more proximal causes of AD. Therefore, *C. elegans* provides a unique system for investigating the mechanisms underlying AD. Besides the usual strengths (e.g. rapid growth cycle, genetic manipulation, cell biological approaches, simple nervous system), *C. elegans* encodes a single APP-like protein, *apl-1*. Unlike APP, APL-1, like its APL-1 mammalian homologs (APLP1 and ALPL2), lack the Abeta peptide sequence. Furthermore, APL-1 lacks the beta-secretase recognition sequences and the *C. elegans* genome does not encode a beta-secretase; therefore, it is unlikely that *C. elegans* produce Abeta peptides (*Daigle and Li, 1993*; *McColl et al., 2012*) and indeed, no Abeta peptides have been detected in *C. elegans* (*McColl et al., 2012*). Thus, *C. elegans* provides a model system that can explore the mechanism underlying AD without the confounding role of Abeta. Excitingly, utilizing this experimental system, we find that in the absence of Abeta, mutations in PSEN result in deregulated ER to mitochondria $Ca^{2+}$ signaling that causes elevated mitochondrial $Ca^{2+}$ which hyperactivates mitochondrial respiration leading to the accumulation of superoxides and the subsequent reduction of neuronal cell function and health.

In spite of our findings indicating that the role of SEL-12 in neurodegeneration is independent of Abeta accumulation, due to the fact that worms do not express Abeta, some toxic effects of Abeta accumulation have been demonstrated in *C. elegans*. Overexpression of human Abeta1-42 results in chemosensory defects, locomotor defects (*Ahmad and Ebert, 2017*; *Dosanjh et al., 2010*; *Fong et al., 2016*; *Wu et al., 2006*) and touch defects (*Figure 5A*). Moreover, similar to the drastic reduction of OXPHOS we observe in older adult *sel-12* mutants compared to age matched wild type animals (*Figure 1—figure supplement 1C,D*), recent studies have shown that overexpression of human Abeta1-42 in the nervous system or body wall muscle results in the reduction of OXPHOS (*Fong et al., 2016*; *Sorrentino et al., 2017*). However, there are some important distinctions between overexpression of human Abeta1-42 and mutations in *sel-12*. Although overexpression of human Abeta1-42 in the nervous system disrupts touch response similar to *sel-12* mutants (*Figure 5A*), it fails to result in hallmark *sel-12* mutation defects such as axonal morphology phenotypes, disorganized mitochondria, and excessive calcium loading (*Figure 5B–E*). Therefore, the mechanism of Abeta1-42 overexpression appears distinct from presenilin mutations. These important findings suggest that an initial insult in FAD, since *C. elegans* does not produce Abeta peptides, is elevated ROS production from the mitochondria due to an increase in ER $Ca^{2+}$ transfer. Thus, Abeta accumulation may be a secondary, albeit critical, component to the pathogenesis of the disease.

## Materials and methods

### Animal maintenance and strains

*C. elegans* are grown on NGM plates between 16–25°C. For all experiments, animals are synchronized by bleaching plates containing gravid animals and the progeny are allowed to hatch in M9 buffer. The synchronized L1 larvae are then grown on NGM plates seeded with OP50 at 20°C, unless otherwise noted, until the necessary stage of growth for each experiment. *glp-1(e2141)* and their wild type counterparts were raised for experiments at 25°C from L1 stage. For experiments that required sterilization, age synchronized L4 animals are moved to 0.5 mg/ml 5-fluorour-aci1-2′-deoxy-ribose (FUDR) containing agar plates seeded with OP50. The following strains were used in this study: Wild type (N2), *sel-12(ar131, ty11 and ok2078) X*, *hop-1(ar179) I*, *crt-1(jh101) V*, *mcu-1 (tm6026) IV*, *glp-1(e2141) III*, *lin-12(ok2215) III*, *unc-68(r1162) V*, CL2355 *smg-1(cc546) dvIs50 [unc-119p::Abeta1-42] I*, *zdIs5 [mec-4p::GFP] I*, *uIs69 [unc-119p::sid-1+myo-2p::mCherry] V*, *jsIs609 [mec-4p::MLS::GFP]*, *zcIs14 [myo-3p::tomm-20::GFP]*, *twnEx8 (mec-7p::tomm-20::mCherry, myo-2p::gfp)*,

*zhsEx17 [Pmec-4mitoLS::ROGFP], takEx214 (myo3p::sel-12::SL2::mCherry; myo2p::mCherry), takEx222 (rab3p::sel-12::SL2::mCherry; ttx3p::GFP).*

## Cell culture

Cell culture media and reagents were purchased from Invitrogen (Waltham, MA) and Corning (Cell-gro, Manassas, VA). FAD (AG07872, AG07768, AG06848, AG08170) and normal control (AG08379, AG07871, AG08701, AG08509) human skin fibroblasts cell lines were obtained from NIA Aging Cell Culture Repository (Coriell, Camden, NJ). Cell lines were authenticated by AmpFLSTR Indentifiler Plus PCR Amplification Kit (ThermoFisher Scientific) and tested for absence of mycoplasma by Myco-SEQ Mycoplasm Detection System (Life Technologies) by Coriell Cell Repositioires. All cell lines were grown at 37°C under humidified air containing 5% CO2. Cells were grown in DMEM. DMEM medium (4.5 mg/L glucose, 110 mg/L sodium pyruvate) was supplemented with penicillin (100 U/ml), streptomycin (100 µg/ml), and 15% fetal bovine serum. Cell culture protocol followed in accordance with those provided by the cell supplier.

## DNA constructs and transgenesis

The *sel-12* genomic fragment was subcloned from F35H12 cosmid (Sanger Center) into pBluescript KS+ using SacI and XhoI. The D226A mutation was generated by site directed mutagenesis using the following primers: 5' GCTGTTTGTTATCTCGGTTTGGGcTCTGGTTGCCGTGCTCACACC 3' and 5' GGTGTGAGCACGGCAACCAGAgCCCAAACCGAGATAACAAACAGC 3'. The final product was confirmed by DNA sequencing. The *myo-3p*::mito-GCaMP6f::SL2::mCherry construct was made by PCR amplification of *myo-3p*::mito-GCaMP6f from pKN#18, which contains *myo-3p*::mito-GCaMP6f and was gene synthesized by Genscript, and SL2::mCherry::*unc-54* 3' UTR from pKN#7, which contains SL2::mCherry::*unc-54* 3' UTR and was gene synthesized by Genscript. These products were combined together using Gibson cloning. The mitochondrial matrix targeting sequence was obtained from the N-terminal cytochrome C oxidase subunit VII (*Akerboom et al., 2013*). The final product was confirmed by DNA sequencing. To generate transgenic animals, Qiagen Midi prepared DNA constructs were injected into N2 animals following standard procedures (*Mello and Fire, 1995*). Transgenic animals were selected using *ttx-3p*::GFP as a marker and appropriate subcellular localization was confirmed by fluorescence microscopy.

## Mitochondrial Ca$^{2+}$ concentration measurement

The basal mitochondrial Ca$^{2+}$ concentration in the body wall muscles and mechanosensory neurons was measured using *takEx347* and *takEx415* animals, respectively expressing *myo-3p*::mito-GCaMP6f::SL2::mCherry and *mec-7p*::mito-GCaMP6f::SL2::mCherry. The fluorescence intensity of both the GCaMP6f (genetically encoded Ca$^{2+}$ indicator) and mCherry (used here as a expression control) were measured using a 10X objective lens on a Zeiss Axio Observer microscope and Andor Clara CCD camera, as previously described (*Sarasija and Norman, 2015*). The GCaMP6 fluorescence intensity in each animal was normalized to mCherry intensity.

## Mechanosensation assay

Animals at day 1, 2 and 3 of adulthood were evaluated for the ability to respond to light touch using an eyebrow hair glued to the end of a Pasteur pipette tip (*Chalfie and Sulston, 1981*). Animals were scored based on their responsiveness to a total of 10 touches; five to the anterior (between the head and vulva) and five touches to the posterior (between the tail and vulva). If the animal moves forwards in response to a light touch in the posterior or backward after an anterior touch, respectively, then the animal receives a score of 1 for a maximum score of 10 (100% response) or a minimum score of 0 (0% response).

## Neuronal and mitochondrial morphology analysis

*zdIs5[Pmec-4::GFP]* and was introduced into various genotypes and was utilized to determine ALM and PLM neuronal morphologies (*Wu et al., 2007*). Age synchronized animals are immobilized on 3% agarose pads using 0.1 µm diameter polystyrene microspheres (Polysciences) and their ALM and PLM neuronal structure is imaged under the 63X oil objective on a Zeiss Axio Observer microscope equipped with a Andor Clara CCD camera. Images were compiled using Metamorph software and

scored positive for ectopic neurite sprouting when a visible GFP-labeled branch is seen stemming from the ALM, presence of wave-like bending, presence of triangular beaded lesions and breaks in the ALM or PLM axons. Mitochondrial structure is analyzed using the *jsIs609 [mec-4p::MLS::GFP]*, which targets GFP to the mitochondria in the mechanosensory neurons (*Fatouros et al., 2012*) or *twnEx8(mec-7p::tomm-20::mCherry)*, which targets mCherry to the outer mitochondrial membrane protein via a fusion with the outer mitochondrial membrane protein, TOMM-20 (*Hsu et al., 2014*). Mitochondrial structures were imaged following the protocol used for neuronal imaging (*Sarasija and Norman, 2018a*). The mitochondria were scored as continuous if they existed as an intact circle, or discontinuous when there were breaks to this circle, under blind conditions.

## Inhibition of gamma-secretase activity

The surface of unseeded NGM plates are coated with 100 µl of 100 µM Compound E (Calbiochem), a gamma-secretase inhibitor and seeded with OP50 the next day. Synchronized L1 staged animals are grown on these plates until day one adults, which are then used for analysis (*Francis et al., 2002*). Gamma-secretase inhibition is confirmed by the *glp-1*-like sterility observed in the *hop-1 (ar179)* mutants that are grown on the Compound E containing plates.

## Antioxidant supplementation

Animals are moved to NGM plates containing 500 µM (2-(2,2,6,6-tetramethylpiperidin-1-oxyl-4-ylamino)−2-oxoethyl) triphenylphosphonium chloride (mitoTEMPO) (Sigma) or 500 µM triphenyl-phosphonium chloride (TPP) (Sigma) as L1 larvae. These animals are then used for touch assay, neuronal sprouting and mitochondrial structure analysis as Day one adults.

## Doxycycline and Ru360 treatment

NGM plates containing 10 µg/ml of doxycycline (Sigma) were seeded with the *E. coli* strain HT115. Gravid adult animals from each strain were bleached and their progeny that were moved onto these plates as eggs were used for experiments as day one adults. Fibroblasts were treated with either 1 µg/ml doxycycline (Sigma) or 10 µM Ru360 (Calbiochem) for 48 hr.

## RNAi treatment

To ensure efficient knockdown in the nervous system we used strains expressing *sid-1* pan-neuronally (*Calixto et al., 2010*). These animals are then age synchronized to larval stage one and then allowed to grow on plates seeded with RNAi bacteria (HT115) carrying either the empty vector or *drp-1 RNAi* (Source BioScience) (*Sarasija and Norman, 2015*). These animals were then used for experiments as young adults.

## ROS measurement

ROS measurements were carried out as previously described (*Sarasija and Norman, 2018b*). In Brief, animals grown on 100 mm NGM plates are washed three times with M9 and gravity separated to remove OP50. Then, the animals are washed once in PBS and resuspended in 100 µl of PBS followed by freeze thawing and 10 s sonication. After spinning down the sample at 20,000 g for 10 min at 4°C, the protein concentration is determined using a BCA protein assay and equal amounts of proteins extract are used for each strain and the samples are incubated with $H_2DCF-DA$ assay (Life Technologies) at 37°C for 24 hr. and a Flex Station 3 Reader (Molecular Devices) is used to measure light intensity. Each sample is measured in duplicates or triplicates and the experiments were repeated three times (three biological replicates).

FAD (AG07872, AG07768, AG06848, AG08170) and normal control (AG08379, AG07871, AG08701, AG08509) human skin fibroblasts were seeded at a density of $1.0 \times 10^5$ cells/well in a 96 well black-walled plate with clear bottoms (Costar, Corning, NY) overnight. Prior to the measurement, media were removed and cells were washed with PBS. Cells were incubated in dark at 37°C with 10 µM $H_2DCF-DA$ solution in PBS per well plates was added for 30 mins. After the incubation period, solution of $H_2DCF-DA$ was removed and cells were suspended in PBS. The fluorescence was measured by Flexstation three microplate reader (ex/em = 485/535 nm). Each sample is measured in triplicates and the experiments were repeated three times (three biological replicates).

## ATP measurement

Animals grown on 100 mm NGM plates are washed three times with M9 and gravity separated to remove OP50. Then, the animals are washed once in TE buffer (100 mM Tris-Cl [pH7.5], 4 mM EDTA) and resuspended in 100 µl of TE buffer followed by freeze thawing, 10 s sonication and boiling for 10 min. After spinning down the sample at 20,000 g for 10 min at 4°C, the protein concentration is determined using a BCA protein assay and ATP concentration is determined as per manufacturer's instruction using the ENLITEN ATP Assay System (Promega, Madison, WI). Each sample is measured in quadruplicates and the experiments were repeated three times (three biological replicates).

ATP release in FAD (AG07872, AG07768, AG06848, AG08170) and normal control (AG08379, AG07871, AG08701, AG08509) human skin fibroblasts was determined using the Enliten ATP assay system (Promega) as described by the manufacturer. The luminescence was measured by integration over a 3 s time interval using the luminometer Flexstation three microplate reader, and normalized to protein content determined by BCA. Each sample is measured in triplicates and the experiments were repeated at least twice (two biological replicates).

## Oxygen consumption rate measurement

Oxygen consumption rate (OCR, indicative of mitochondrial OXPHOS) in day one sterilized animals was measured using the Seahorse XFp Extracellular Flux Analyzer (Agilent Seahorse Technologies) adapting the protocol used to measure OCR in *C. elegans* using a Seahorse XF96 Extracellular Flux Analyzer (*Koopman et al., 2016*). Each assay run compared the OCR between two genotypes, in three wells each and five measurements were made for each well for basal OCR and maximal OCR upon addition of FCCP. Assays were repeated on three separate days and three biological replicates were analyzed.

Mitochondrial OXPHOS in human fibroblasts (AG08379 and AG06848) was analyzed using a Seahorse XFp Extracellular Flux Analyzer (Agilent Seahorse Technologies) by measuring the OCR in real time. For OCR analysis, cells were seeded in 8-well plates designed for XFp at 25,000 cells per well in complete growth media. On the next day, the cells were switched to unbuffered media (supplemented with 2.5 mM glucose, 1 mM pyruvate, and 1 mM glutamine) and further incubated in a $CO_2$-free incubator for 1 hr prior to measurement. During measurement, oligomycin (1 µM), FCCP (1 µM), antimycin A and rotenone (0.5 µM) were added. At the end of each assay, protein quantification was performed for normalization. All experiments were performed in triplicate and repeated three times.

## Mitochondrial ROS measurement

Age synchronized young adult animals carrying *zhs17 [Pmec-4mitoLS::ROGFP]* with or without treatment (MitoTEMPO or TPP) were immobilized on 5% agarose pad using 2.4 mg/ml solution of levamisole. They were then imaged on an Olympus Fluoview 1200 Laser Scanning Confocal Microscope with a 60X oil immersion objective lens. Samples were first excited with a 10% power 405 nm laser and then a 10% power 488 nm laser with sequential scanning method and a 0.5 um step size and GFP emission was detected. Images were quantified using ImageJ and the ratio of 405 nm channel was divided by the 488 nm channel.

## Abeta western analysis

Animals were washed off with M9 and gravity separated to remove OP50. This is followed by three washes with dH20 and animals were resuspended in a small volume of RIPA buffer with protease inhibitor cocktail (Roche Diagnostic). Animals were flash frozen at −80C, sonicated and spun down to obtain animal pellet and lysate. The lysates were used to determine protein concentration. The worm pellets were boiled in SDS loading buffer containing 2-beta mercaptoethanol and 60 µg of protein (determined from the protein concentration of the lysate) was loaded in each well of a 12% acrylamide SDS-Page gel. The proteins were transferred to 0.2 µm nitrocellulose membrane and stained with ponceau to ensure equal loading. The nitrocellulose blots were then boiled in PBS, washed in TBS-T and blocked with 5% milk in TBS-T. The blots were incubated overnight in primary antibody against Abeta (6E10, Biolegend; 1:1000) and then with secondary antibody (anti-mouse

IgG (whole molecule)-peroxidase, Sigma-Aldrich, 1:20,000), after washes with TBS-T. The proteins were visualized using Clarity TM Western ECL Substrate (BIO-RAD) and X-ray film.

### Statistical analysis

GraphPad Prism software is used for statistical analysis. Student's t test is used only for comparing two samples and one-way ANOVA with Tukey post test has been used when making multiple comparisons. Chi-square test was used to compare the difference in mitochondrial structural phenotypes between strains.

## Acknowledgements

We thank Monica Driscoll, Natalia Morsci, Chun-Liang Pan, and Caenorhabditis Genetics Center [supported by National Institutes of Health – Office of Research Infrastructure Programs (P40 OD010440)] for strains. National Institutes of Health grant R01GM088213 supported this work.

## Additional information

### Funding

| Funder | Grant reference number | Author |
| --- | --- | --- |
| National Institute of General Medical Sciences | R01 GM088213 | Kenneth R Norman |

The funders had no role in study design, data collection and interpretation, or the decision to submit the work for publication.

### Author contributions

Shaarika Sarasija, Conceptualization, Data curation, Formal analysis, Validation, Investigation, Visualization, Methodology, Writing—original draft, Writing—review and editing; Jocelyn T Laboy, Resources, Data curation, Methodology; Zahra Ashkavand, Data curation, Formal analysis, Investigation; Jennifer Bonner, Resources, Formal analysis, Investigation, Methodology; Yi Tang, Resources, Methodology; Kenneth R Norman, Conceptualization, Resources, Data curation, Formal analysis, Supervision, Funding acquisition, Investigation, Visualization, Methodology, Writing—original draft, Project administration, Writing—review and editing

### Author ORCIDs

Shaarika Sarasija https://orcid.org/0000-0002-3610-2178
Kenneth R Norman http://orcid.org/0000-0002-0773-9073

### Decision letter and Author response

Decision letter https://doi.org/10.7554/eLife.33052.041
Author response https://doi.org/10.7554/eLife.33052.042

## Additional files

### Supplementary files

• Transparent reporting form
DOI: https://doi.org/10.7554/eLife.33052.039

### Data availability

Source data files have been provided for all figures.

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
