## [Decision Letter]

Thank you for submitting your article "Presenilin mutations deregulate mitochondrial Ca^2+^ homeostasis and metabolic activity causing neurodegeneration" for consideration by *eLife*. Your article has been reviewed by two peer reviewers, and the evaluation has been overseen by a Reviewing Editor and a Senior Editor. The reviewers have opted to remain anonymous.

The reviewers have discussed the reviews with one another and the Reviewing Editor has drafted this decision to help you prepare a revised submission. While there is enthusiasm for this body of work, there is also concern about the completion of the outlined experiments.

The authors of the study propose that neurodegeneration occurring in Alzheimer's disease might be completely independent of amyloid aggregates presence, and they attempt to demonstrate this hypothesis using mostly *C. elegans* presenilin mutants. These should offer the advantage of a clean model where to study the effects of LOF presenilin on neurodegeneration independently of Abeta amyloid generation. They show that in the presenilin mutants neurodegeneration occurs by faulty ER to mitochondria Ca^2+^ transfer, which leads to increased mitochondrial ROS generation and conclude that Abeta presence is not necessary for neurodegeneration.

There are two major lines of inquiry that need to be addressed before this paper can be considered for publication in *eLife*. One, the authors need to firmly address whether increased OXPHOS and ATP production and increased Ca^2+^ is indeed the culprit behind the increased respiration rates. Two, the results surrounding the PSEN mutation in the absence of Abeta production are bold, yet lack experimentation. Within both lines of query, the worm model can be used and the reviewers are not asking for new discoveries in vertebrate models, but rather formal testing in existing worm models. A detailed list of experiments is bullet pointed to guide the authors.

1) Does ETC respiration play a direct role on the phenotypes observed? In worms this is simple to achieve with RNAi towards any one of the ETC subunits, mitochondrial ribosome components or merely treating with doxycycline. This needs to also be confirmed in the human patient cell lines.

2) Figure 2 describes defects in mito morphology, fragmented, that do not really fit the OCR and OXPHOS results. The authors need to block fission and test if the *sel-12* mutant is still effective. RNAi towards *drp-1* should suffice.

3) The calreticulin mutant experiment is over interpreted on its own. The loss of *crt-1* will indeed disrupt Ca^2+^ signaling from the ER, but it also creates a mess for the ER affecting many other things. Therefore, I think another approach should be employed and OCR and OXPHOS on these animals must be performed.

4) The MitoTEMPO experiments are neat, but uncontrolled. The authors need to test ROS levels in the MitoTEMPO *sel-12* mutants as well as express a catalytically dead version of MitoTEMPO to ensure that it is not merely affecting Ca^2+^ levels on its own. Therefore, Ca^2+^ levels need to be measured as well.

5) The authors insist that Abeta peptide production is not necessary for neurodegeneration. However, existence of amyloid plaque accumulation is a well-known pathological hallmark of AD and using only *C. elegans* devoid of amyloid plaque accumulation as an AD model is hence not sufficient for such a bold conclusion. The authors should measure neurodegeneration and mitochondrial Ca^2+^ levels in a neuronal/muscle Abeta expressing worm line. In case these worms have Ca^2+^ levels perturbations, they should try to rescue neurodegeneration by normalizing mitochondrial Ca^2+^ levels. Moreover, they should clearly state that their conclusions are a) limited to this model system, and b) show that mutations in PSEN are enough to cause neurodegeneration in *C. elegans*, but one can't exclude the additional contributions of Abeta in humans.

6) A recent paper studying MEFs from PSEN KO mice (Contino et al., 2017), shows results that seems to contradict the result presented in this manuscript, with mitochondria showing decreased OXPHOS, respiration and ATP production in MEFs KO cells. How could the authors reconcile their proposed mechanism with this work?

[Editors' note: further revisions were requested prior to acceptance, as described below.]

Thank you for resubmitting your work entitled "Presenilin mutations deregulate mitochondrial Ca^2+^ homeostasis and metabolic activity causing neurodegeneration in *Caenorhabditis elegans*" for further consideration at *eLife*. Your revised article has been favorably evaluated by a Senior Editor and Reviewing Editor.

In the revised manuscript, the authors have performed a number of additional experiments. The revised manuscript is thereby largely improved, however, there are still a few aspects that require clarity in the final manuscript.

1) The doxycycline experiments show a clear improvement of the phenotypes in worms and cells. However, while for worms the authors use an amount of compound in the range of what also reported by literature (μg/ml), for the fibroblast experiments they report a concentration of 5 mg/ml, which seems very high. Lower concentrations (in the 0.5-30 μg/ml range) have been shown to affect mitochondrial function in cells, while still not affecting viability, why is such a high amount used here? Are the cells still viable? An experiment with a lower dosage of dox, or a dose titration study, should be included.

2) While the authors have performed the Abeta expression experiments in the worms, the description and experiments fully proving the presence and effect of Abeta are still superficial. First, a detailed description of the methods used to generate the transgenic Abeta worms should be included. Secondly, experiments showing the presence and expression of Abeta peptide should be included. Third, while Ca^2+^ levels and mitochondrial morphology do not seem affected, literature has largely described that the presence of Abeta in worms' neurons (one example, doi:10.1038/srep33781) or in muscle (for instance, doi: 10.1038/nature25143) consistently affects mitochondrial function by reducing respiration and affecting other mitochondrial parameters. The authors need to bring these papers into their Discussion.

---

## [Author Response]

[…] There are 2 major lines of inquiry that need to be addressed before this paper can be considered for publication in eLife. One, the authors need to firmly address whether increased OXPHOS and ATP production and increased Ca^2+^ is indeed the culprit behind the increased respiration rates. Two, the results surrounding the PSEN mutation in the absence of Abeta production are bold, yet lack experimentation. Within both lines of query, the worm model can be used and the reviewers are not asking for new discoveries in vertebrate models, but rather formal testing in existing worm models. A detailed list of experiments is bullet pointed to guide the authors.1) Does ETC respiration play a direct role on the phenotypes observed? In worms this is simple to achieve with RNAi towards any one of the ETC subunits, mitochondrial ribosome components or merely treating with doxycycline. This needs to also be confirmed in the human patient cell lines.

To address this concern, we took the reviewers’ suggestion and inhibited ETC by adding an appropriate amount of doxycycline to our worms or cultured fibroblasts. From these analyses, we found mitochondrial ETC respiration has a critical role in the *sel-12* mutant neurodegenerative phenotype. This data is incorporated into a new figure (Figure 8) and the Results section (subsection “Reducing mitochondrial function but not mitochondrial fission rescues neurodegeneration in *sel-12* mutants”).

2) Figure 2 describes defects in mito morphology, fragmented, that do not really fit the OCR and OXPHOS results. The authors need to block fission and test if the sel-12 mutant is still effective. RNAi towards drp-1 should suffice.

To address this concern, we again listened to the reviewers suggestions and used *drp-1(RNAi)* to improve mitochondrial structure in *sel-12* mutants and analyzed mitochondrial function. From these analyses, while we could improve mitochondrial morphology, we did not see an improvement in the *sel-12* phenotype. These data are presented in a new figure (Figure 8—figure supplement 1) and a new Results subsection “Reducing mitochondrial function but not mitochondrial fission rescues neurodegeneration in *sel-12* mutants”.

3) The calreticulin mutant experiment is over interpreted on its own. The loss of crt-1 will indeed disrupt Ca^2+^ signaling from the ER, but it also creates a mess for the ER affecting many other things. Therefore, I think another approach should be employed and OCR and OXPHOS on these animals must be performed.

The other approach we used to address this concern was to introduce an *unc-68* null mutation (*unc-68* encodes the sole ryanodine receptor in the *C. elegans* genome) in the *sel-12* background and tested whether loss of *unc-68* could suppress the elevated mitochondrial Ca^2+^ levels, abnormal axon morphology and elevated OCR we observe in *sel-12* mutants. Similar to the *crt-1* null mutation, we found that *unc-68* loss could also suppress the *sel-12* phenotypes. This data is shown in a new figure (Figure 6—figure supplement 1) and in the Results (subsection “Decreasing ER Ca^2+^ release improves mitochondrial function and suppresses neurodegeneration in *sel-12* mutants”.

4) The MitoTEMPO experiments are neat, but uncontrolled. The authors need to test ROS levels in the MitoTEMPO sel-12 mutants as well as express a catalytically dead version of MitoTEMPO to ensure that it is not merely affecting Ca^2+^ levels on its own. Therefore, Ca^2+^ levels need to be measured as well.

To address these concerns, we tested whether MitoTEMPO could reduce mitochondrial Ca^2+^ levels and/or ROS levels. To accomplish this we utilized our mGCaMP6/mCherry construct and measured the ratio of GCaMP6/mCherry fluorescence in the presence or absence of MitoTEMPO. Similarly, we utilized a redox sensitive GFP (roGFP) that is expressed in the mechanosensory neurons to gauge whether MitoTEMPO is reducing ROS in the mitochondria of these neurons. We found that MitoTEMPO did not influence mitochondrial Ca^2+^ levels but did reduce mitochondrial ROS levels. These data are presented in the subsection “Mitochondrial generated superoxides cause mechanosensation and neuronal morphology defects in *sel-12* mutants” and in Figure 9A, Figure 9—figure supplement 1A and B. Also, all experiments were repeated in the presence of TPP. These data are included in the aforementioned subsection and figure supplements and in Figure 9B and C.

5) The authors insist that Abeta peptide production is not necessary for neurodegeneration. However, existence of amyloid plaque accumulation is a well-known pathological hallmark of AD and using only C. elegans devoid of amyloid plaque accumulation as an AD model is hence not sufficient for such a bold conclusion. The authors should measure neurodegeneration and mitochondrial Ca^2+^ levels in a neuronal/muscle Abeta expressing worm line. In case these worms have Ca^2+^ levels perturbations, they should try to rescue neurodegeneration by normalizing mitochondrial Ca^2+^ levels. Moreover, they should clearly state that their conclusions are a) limited to this model system, and b) show that mutations in PSEN are enough to cause neurodegeneration in C. elegans, but one can't exclude the additional contributions of Abeta in humans.

To address this concern, we have expressed human Abeta1-42 pan-neuronally and examined light touch response, mitochondrial Ca^2+^ levels, and mitochondria and axon morphology. From these analyses, we found that touch response in animals overexpressing Abeta showed a similar defect as observed in *sel-12* mutants. However, unlike *sel-12* mutants, the mitochondrial Ca^2+^ levels and mitochondrial morphology are normal. Moreover, axon morphology was normal in Abeta overexpressing animals. These data are presented in the subsection “Expression of human Abeta1-42 peptide results in loss of mechano-perception in a mechanism disparate from loss of SEL-12” and in Figure 5.

6) A recent paper studying MEFs from PSEN KO mice (Contino et al., 2017), shows results that seems to contradict the result presented in this manuscript, with mitochondria showing decreased OXPHOS, respiration and ATP production in MEFs KO cells. How could the authors reconcile their proposed mechanism with this work?

While the data from Contino et al. appears to contradict our data to a certain degree, there are some significant differences. 1) Their data is collected from mouse embryonic fibroblasts, 2) their culture conditions are in low glucose and it is unclear what the glucose (or pyruvate or glutamine) concentration was used for their OCR recordings, and 3) they find that presenilin 2 has a role in mitochondrial function unlike our studies in skin fibroblasts isolated from FAD patient who have mutations in the gene encoding presenilin 1. Moreover, a recent study published in the Stem Cell Reports by Oksanen et al., (Stem Cell Reports 2017) examining PSEN1 iPSCs generated from patients with a mutation in presenilin 1 show elevated OCR and high ROS levels, similar to what we observe in FAD skin fibroblasts and *C. elegans sel-12* mutants. Additionally, these authors found that CRISPR/Cas9 mediated correction of the PSEN1 mutation in these iPSCs could reduce the OCR and ROS levels, thereby, implicating PSEN1 in mitochondrial function. We have discussed these points in the subsection “Skin fibroblasts isolated from FAD patients have increased ATP, OCR and ROS levels” and Discussion, third paragraph.

[Editors' note: further revisions were requested prior to acceptance, as described below.]

In the revised manuscript, the authors have performed a number of additional experiments. The revised manuscript is thereby largely improved, however, there are still a few aspects that require clarity in the final manuscript.1) The doxycycline experiments show a clear improvement of the phenotypes in worms and cells. However, while for worms the authors use an amount of compound in the range of what also reported by literature (μg/ml), for the fibroblast experiments they report a concentration of 5 mg/ml, which seems very high. Lower concentrations (in the 0.5-30 μg/ml range) have been shown to affect mitochondrial function in cells, while still not affecting viability, why is such a high amount used here? Are the cells still viable? An experiment with a lower dosage of dox, or a dose titration study, should be included.

For the fibroblast experiments, we used 1μg/mL as described in Moullan et al., 2015. We appreciate and thank the reviewers and editor for catching this mistake. This has been corrected in the subsection “Doxycycline and Ru360 Treatment”.

2) While the authors have performed the Abeta expression experiments in the worms, the description and experiments fully proving the presence and effect of Abeta are still superficial. First, a detailed description of the methods used to generate the transgenic Abeta worms should be included. Secondly, experiments showing the presence and expression of Abeta peptide should be included. Third, while Ca^2+^ levels and mitochondrial morphology do not seem affected, literature has largely described that the presence of Abeta in worms' neurons (one example, doi:10.1038/srep33781) or in muscle (for instance, doi: 10.1038/nature25143) consistently affects mitochondrial function by reducing respiration and affecting other mitochondrial parameters. The authors need to bring these papers into their Discussion.

We address each point below:

1) The Abeta strain used for this study is available from the Caenorhabditis Genetics Center and has been characterized by multiple labs and several publications utilizing this reagent exist in the literature (e.g. Wu et al., 2006; Dosanjh et al., 2010; Van Assche et al., 2015 doi: 10.1007/s11306-014-0711-5; Zhang et al., 2016 doi: 10.1016/j.jep.2016.09.031; Jan et al., 2017 doi: 10.1007/s00401-016-1634-1; Ahmad and Ebert, 2018 doi: 10.1016/j.exger.2018.04.021; VazBravo et al., 2018 doi: 10.1038/s41598-018-21918-5). The strain is listed in the “Animal maintenance and strains” subsection (CL2355). It was raised at 20°C for analysis as described in the animal maintenance and strain section. The phenotype analyses were carried out as described in the Materials and methods.

2) While presence and expression has been shown previously (Wu et al., 2006; Dosanjh et al., 2010), we have conducted western analysis to authenticate that this reagent expresses and we can detect the presence of Abeta under the conditions we used for our analyses. This data is included in Figure 5—figure supplement 1 and in the subsection “Expression of human Abeta1-42 peptide results in loss of mechano-perception in a mechanism disparate from loss of SEL-12”. Methods have been added subsection “Abeta western analysis”.

3) This is one of the exciting points of our manuscript. While the Abeta hypothesis is a large driving force in AD research, there are many studies that suggest Abeta may be a secondary insult in AD pathogenesis (e.g. Herrup 2015 doi: 10.1038/nn.4017). In our study, we demonstrate that mutations in presenilin result in defects in ER-mitochondrial calcium handling that leads to elevated OXPHOS and ROS generation. This causes neurodegenerative phenotypes. While it is clear from others and in our own work that ectopic overexpression of Abeta is toxic and can lead to defects in cellular function, survival, and mitochondrial activity, the mechanism appears distinct from presenilin mutations. This important finding suggests that the initial insult in FAD, since *C. elegans* does not produce Abeta peptides, is elevated ROS production from the mitochondria due to an increase in ER Ca^2+^ transfer. Thus, Abeta accumulation may be merely secondary, albeit critical to the pathogenesis of the disease.

Importantly, our study focuses mainly on the loss or reduction of function mutations in *sel-12* and not the ectopic overexpression of an exogenous protein, which is well known to be prone to aggregation and toxic. Moreover, our study of *sel-12* mutants shows neuronal defects in the absence of Abeta peptides, which provides evidence that PSEN dysfunction is likely an initial cause of neurodegeneration and Abeta peptide aggregation may be a secondary insult.

We have discussed our results in relation to the studies by Fong et al., 2016 and Sorrentino et al., 2017 at the end of the Discussion.